# Asymmetrical deposition and modification of histone H3 variants are essential for zygote development

Machika Kawamura, Satoshi Funaya, Kenta Sugie, Masataka G Suzuki, Fugaku Aoki

The pericentromeric heterochromatin of one-cell embryos forms a unique, ring-like structure around the nucleolar precursor body, which is absent in somatic cells. Here, we found that the histone H3 variants H3.1 and/or H3.2 (H3.1/H3.2) were localized asymmetrically between the male and female perinucleolar regions of the one-cell embryos; moreover, asymmetrical histone localization influenced DNA replication timing. The nuclear deposition of H3.1/3.2 in one-cell embryos was low relative to other preimplantation stages because of reduced H3.1/3.2 mRNA expression and incorporation efficiency. The forced incorporation of H3.1/3.2 into the pronuclei of one-cell embryos triggered a delay in DNA replication, leading to developmental failure. Methylation of lysine residue 27 (H3K27me3) of the deposited H3.1/3.2 in the paternal perinucleolar region caused this delay in DNA replication. These results suggest that reduced H3.1/3.2 in the paternal perinucleolar region is essential for controlled DNA replication and preimplantation development. The nuclear deposition of H3.1/3.2 is presumably maintained at a low level to avoid the detrimental effect of K27me3 methylation on DNA replication in the paternal perinucleolar region.

## Introduction

Heterochromatin, which is defined as transcriptionally inert and condensed chromatin, is heavily involved in the regulation of cellular processes such as gene expression, mitosis, and DNA replication (Campos & Reinberg, 2009; Saksouk et al, 2015). Constitutive heterochromatin is relatively gene-poor and mainly composed of tandem satellite repeats. It is present in the pericentromeric, telomeric, and ribosomal regions of all cell types (Saksouk et al, 2015). Pericentromeric heterochromatin can be identified microscopically as foci within the nuclear region that is densely stained with 4′,6-diamidino-2-phenylindole (DAPI). In these regions, satellite repeats are transcriptionally silenced and DNA is late-replicating at the S phase (O'Keefe et al, 1992; Probst & Almouzni, 2011; Saksouk et al, 2015). However, the characteristics of pericentromeric heterochromatin differ in one-cell-stage embryos,

where it forms a ring-like structure around the nucleolar precursor body; this is referred to as the perinucleolar region (Akiyama et al, 2011). Here, the satellite repeats are actively transcribed, and the timing and sequence of DNA replication differ from those aspects in somatic cells (O'Keefe et al, 1992; Ferreira & Carmo-Fonseca, 1997; Aoki & Schultz, 1999). Notably, temporal differences in cellular processes between maternal and paternal pronuclei have been observed in the perinucleolar region. Transcriptional activity of satellite repeats is higher and DNA replication is completed earlier, in paternal pericentromeric heterochromatin than in maternal heterochromatin (Aoki & Schultz, 1999; Puschendorf et al, 2008; Probst et al, 2010; Santenard et al, 2010). These differences suggest that the chromatin structure of the paternal perinucleolar region forms a looser chromatin structure, compared with the maternal perinucleolar region. However, the mechanisms driving the structural and process-related differences between parental nuclei in pericentromeric heterochromatin have not been well characterized.

Recent studies have revealed epigenetic asymmetry between the maternal and paternal pronuclei. Pericentromeric heterochromatin is similar between the maternal perinucleolar region and somatic cells, such that it contains histone H3 di/trimethylated at lysine 9 (H3K9me2/3) and H4 trimethylated at lysine 20 (H4K20me3) (Lepikhov & Walter, 2004; Santos et al, 2005; Probst et al, 2007; Puschendorf et al, 2008). However, the pericentromeric heterochromatin in the paternal pronucleus lacks these typical heterochromatin modifications (Probst et al, 2010); instead, it contains H3 trimethylated at lysine 27 (H3K27me3) and H2A ubiquitylated at lysine 119 (H2AK119ub) (Puschendorf et al, 2008; Tardat et al, 2015; Eckersley-Maslin et al, 2018). In addition, heterochromatin protein 1 is recruited to the maternal perinucleolar region, whereas polycomb repression complexes 1 and 2 regulate the paternal perinucleolar region (Tardat et al, 2015). However, the contribution of these epigenetic factors to the asymmetry of cellular processes between parental pericentromeric regions has not been assessed thus far.

Histone variants are key factors determining chromatin structure. Several recent studies have focused on histone H3 variants, which share highly similar amino acid sequences but display distinctive characteristics and functions. In mammals, there are three non-centromeric histone variants: H3.1, H3.2, and H3.3 (Hake & Allis, 2006). H3.1 and H3.2 are expressed and incorporated into chromatin in a DNA replication–dependent manner (Tagami et al,

Department of Integrated Biosciences, Graduate School of Frontier Sciences, The University of Tokyo, Kashiwa, Japan

Correspondence: aokif@edu.k.u-tokyo.ac.jp

2004; Hake & Allis, 2006). As revealed by ChIP-seq analyses using FLAG-tagged histone variants expressed in embryonic stem cells, H3.1 and H3.2 are generally deposited in both euchromatin and heterochromatin (Yukawa et al, 2014). H3.3 is expressed and incorporated into chromatin in a DNA replication–independent manner (Hake & Allis, 2006). H3.3 is generally incorporated into euchromatic regions. However, it has recently been revealed that H3.3 also localizes to pericentromeric repeats (Rapkin et al, 2015), suggesting that H3.3 can be incorporated into heterochromatin. In one-cell embryos, H3.3 is incorporated in the paternal perinucleolar region by the recruitment of DAXX, a chaperone of H3.3, which is mediated by PRC1 containing SUMOylated CBX2 (Santenard et al, 2010; Liu et al, 2020). This H3.3 incorporation in the paternal perinucleolar region is suggested to regulate the transcription of major satellite repeats (Santenard et al, 2010) and the formation of compact heterochromatin (Liu et al, 2020). Therefore, H3.1, H3.2, and H3.3 have distinct characteristics and mechanisms of chromatin incorporation; they may contribute to regulation of cellular process asymmetry that occurs in parental perinucleolar regions.

In this study, we investigated the involvement of histone H3 variants in structural and cellular process asymmetry in the pericentromeric heterochromatin between the parental genomes of mouse embryos at the one-cell stage. We found that H3 variants were localized asymmetrically between the maternal and paternal perinucleolar regions. The maternal and paternal perinucleolar regions were enriched in H3.1/2 with K9me2/3 (H3.1/2K9me2/3) and H3.3 with K27me3 (H3.3K27me3), respectively. The forced incorporation of H3.1 and H3.2 into the paternal pronucleus caused an increase in H3.1/2K27me3 and a delay in DNA replication in the perinucleolar region, leading to developmental failure. These results suggest that the nuclear configuration of H3 variants causes the asymmetric chromatin structure in parental pronuclei, and that reduced H3.1/2 nuclear deposition in the paternal perinucleolar region prevents accumulation of H3.1/2K27me3 modification, which has a detrimental effect on DNA replication.

# Results

## Nuclear deposition of histone H3 variants in preimplantation embryos

Using immunocytochemical techniques, we investigated the nuclear deposition of H3 variants using antibodies that recognize both H3.1 and H3.2 (H3.1/2), or H3.3. The former antibody does not discriminate between H3.1 and H3.2. The specificity of the antibodies used was verified by antigen peptide adsorption (Fig S1). The H3.3 signal was clearly detected in the nuclei throughout each stage of preimplantation development (Fig 1A), which was consistent with previous reports (Torres-Padilla et al, 2006; Akiyama et al, 2011). However, the H3.1/2 signal was nearly absent in the pronuclei of one-cell embryos. H3.1/2 was detected in the nuclei of two-cell-stage embryos; the H3.1/2 signal increased at the four-cell stage. Inhibition of DNA replication by treatment of embryos with aphidicolin prevented the nuclear deposition of H3.1/2 in two-cell-stage embryos (Fig S2A), indicating that H3.1/2 is deposited into chromatin

in a DNA replication–dependent manner during the two-cell stage, as in somatic cells (Tagami et al, 2004).

Although H3.1/2 was not detected in the pronuclei of one-cell embryos in the initial observation, it was clearly visible when the confocal laser scanning microscope detector gain was enhanced (Fig 1B). The pronuclear deposition of H3.1/2 is also DNA replication–dependent at both one-cell and two-cell stages because treatment with aphidicolin inhibited their H3.1/2 signals (Fig S2A and B). Notably, the patterns of H3.1/2 localization differed between parental pronuclei. Although the signal intensity of H3.1/2 was similar between the two pronuclei in the nucleoplasm, the signal was more intense in the perinucleolar region of the maternal pronucleus than in the paternal perinucleolar region. These results suggest that the composition of histone variants constituting pericentromeric heterochromatin differs between maternal and paternal pronuclei because the pericentromeric heterochromatin is localized to the rim of pronucleoli at the one-cell stage (Ferreira & Carmo-Fonseca, 1997; Akiyama et al, 2011). Previous reports showed that from the late two-cell stage onwards, pericentromeric heterochromatin forms chromocenters, which appear under the microscope as foci of high DNA density (Martin et al, 2006). H3.1/2 colocalized with these chromocenters in two-cell-stage embryos (Fig 1C), which suggested that H3.1/2 is involved in the formation of pericentromeric heterochromatin in early preimplantation embryos.

## Limited H3.1/2 nuclear deposition in one-cell embryos is caused by low H3.1/2 mRNA expression and incorporation efficiency

There are two possible mechanisms that cause the limited nuclear deposition of H3.1/2 in one-cell embryos: low H3.1/2 expression and/or reduced incorporation efficiency of H3.1/2 into chromatin. To identify which mechanism participates in this process, we analyzed the mRNA levels of H3.1, H3.2, and H3.3 using previously published RNA-seq data (Abe et al, 2015). Histone H3 variants are encoded by multiple genes. In mice, H3.1, H3.2, and H3.3 are encoded by four, eight, and two genes, respectively (Wang et al, 1996a, 1996b; Tang et al, 2015). The reads per kilobase of exon per million mapped reads (RPKM) values for each gene were added to calculate the total RPKM value for each H3 variant. The RPKM value of H3.3 at the one-cell stage was normalized to 1 and the relative RPKM values for H3.1/2 variants were calculated (Fig 2A). In the stages at which H3.1/2 nuclear deposition was clearly detected (i.e., two-cell stage onwards; Fig 1A), the relative RPKM values for H3.1 and H3.2 were equivalent to (or higher than) the relative RPKM value of H3.3. However, at the one-cell stage, in which the level of H3.1/2 nuclear localization was low, the relative RPKM value of H3.1/2 was lower than the relative RPKM value of H3.3. These results suggested that low levels of H3.1/2 mRNA contribute to reduced pronuclear H3.1/2 deposition at the one-cell stage.

To compare the incorporation efficiencies of H3.1 and H3.2 relative to H3.3, C-terminally FLAG-tagged cRNA encoding H3.1, H3.2, or H3.3 was microinjected into metaphase II (MII)-stage oocytes at various concentrations (3, 10, 30, and 100 ng/μl). The oocytes were inseminated and collected for immunocytochemical analysis with an anti-FLAG antibody at 11 h post-insemination (hpi). Quantification of anti-FLAG signal intensities revealed that the incorporation efficiencies of H3.1 and H3.2 were significantly lower than those of H3.3 at cRNA concentrations of ≤30 ng/μl (Fig 2B). Notably, the incorporation efficiencies of the three H3

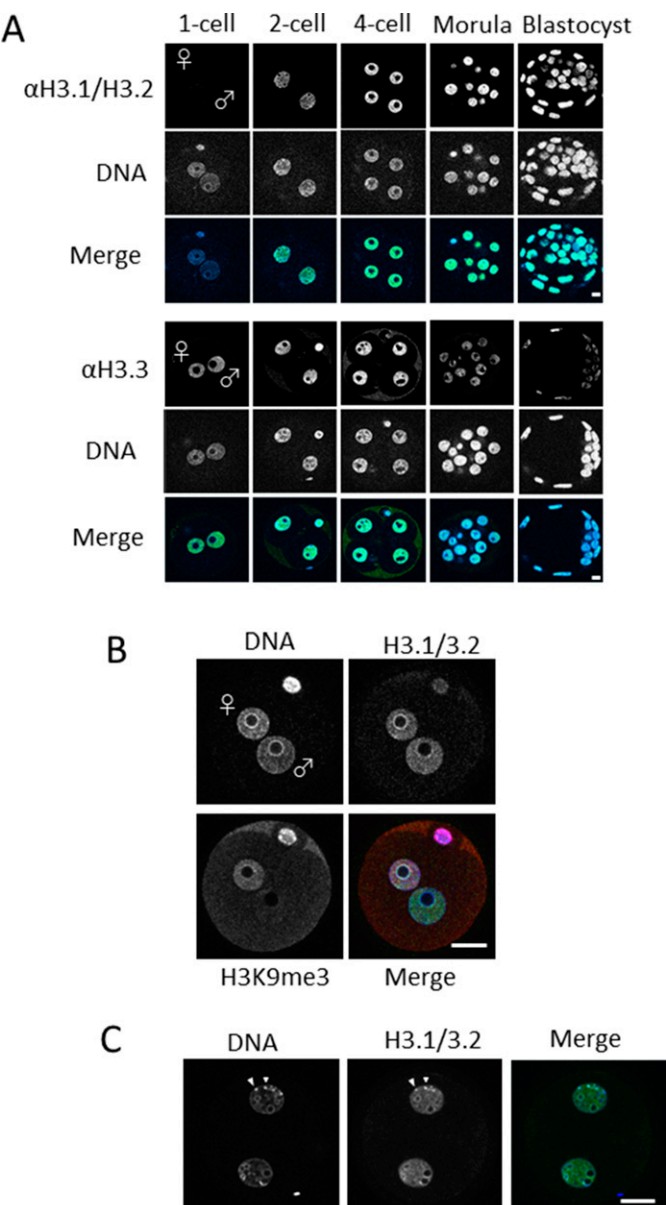

**Figure 1. Nuclear deposition of histone H3 variants in mouse preimplantation embryos.**
**(A)** One-cell, two-cell, four-cell, morula, and blastocyst-stage embryos were immunostained using anti-H3.1/2 (top half) and anti-H3.3 (bottom half) antibodies. Four to five independent experiments were performed. 8–15 embryos were observed for each developmental stage in each experiment; 39–64 embryos were analyzed in total. Representative images are shown for each experiment. Scale bar, 10 μm. **(B)** Enlarged images of stained one-cell embryos with enhanced confocal detector gain. In addition to H3.1/2, H3K9me3 was immunostained to discriminate the male and female pronucleus. In the merged panel, blue, green, and red colors represent the signals of DNA, H3.1/2, and H3K9me3, respectively. Scale bar, 20 μm. **(C)** Enlarged images of two-cell embryos; arrowheads indicate chromocenters. In the merged panel, blue and green colors represent the signals of DNA and H3.1/2, respectively. Scale bar, 20 μm.

variants were similar when 100 ng/μl cRNA was microinjected. These findings indicate that the relatively low nuclear localization of H3.1 and H3.2 in one-cell embryos is caused by both reduced H3.1/2 mRNA expression and low incorporation efficiency into chromatin, relative to H3.3.

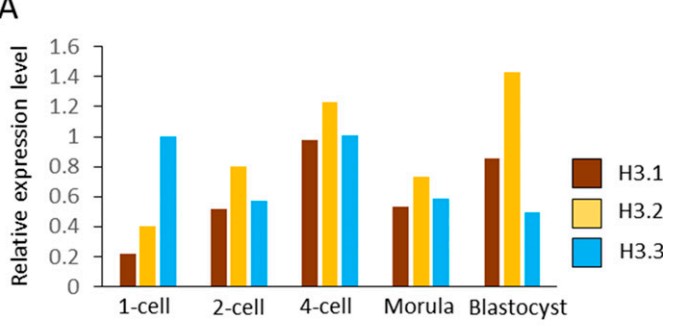

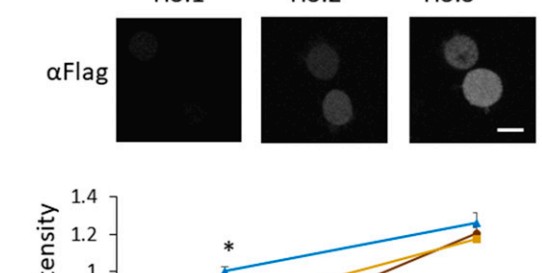

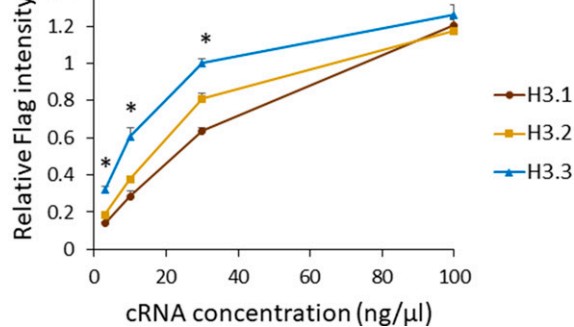

**Figure 2. Nuclear localization of H3.1/2 is regulated by mRNA levels and histone incorporation efficiency in one-cell-stage embryos.**
**(A)** mRNA expression levels of H3 variants during preimplantation development. RPKM values were obtained from previously published RNA-seq data (Abe et al, 2015). RPKM values for each gene encoding H3.1, H3.2, or H3.3 were totaled; the total RPKM of H3.3 at the one-cell stage was normalized to 1. **(B)** The incorporation efficiency of histone H3 variants into chromatin of one-cell embryos. Approximately 10 pl of H3.1, H3.2, or H3.3-FLAG cRNA was microinjected into MII-stage oocytes at various concentrations (3, 10, 30, and 100 ng/μl). After insemination, embryos were collected at 11 h post-insemination (hpi) and subjected to immunostaining. Anti-FLAG antibody was used to detect FLAG-tagged histones incorporated into chromatin. Representative immunocytochemistry images depict one-cell embryos, in which 10 ng/μl of H3.1, H3.2, or H3.3-FLAG was microinjected. Scale bar, 10 μm. The incorporation efficiency of H3 variants at one-cell embryos is shown as a line graph. The signal intensity for H3.3 microinjected at 30 ng/μl concentration was normalized to 1. Nine experiments were performed in total, using H3.3 injected with 30 ng/μl as a control for each experiment. Three to four experiments were performed for each concentration. Ninety 1-cell embryos were analyzed for the H3.3 30 ng/μl concentration. For embryos microinjected with other cRNA concentrations, 26–43 embryos were analyzed in total. Bars indicate standard error. Asterisks indicate that the value for H3.3 was significantly higher than both values of H3.1 and H3.2 (P < 0.01 by t test).

## Biological significance of limited H3.1/2 nuclear deposition in one-cell embryos

To examine the biological significance of the low H3.1/2 levels in one-cell embryos, we forced the incorporation of H3.1/2 into

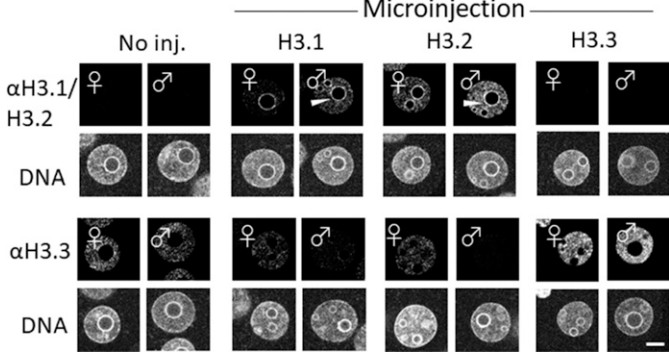

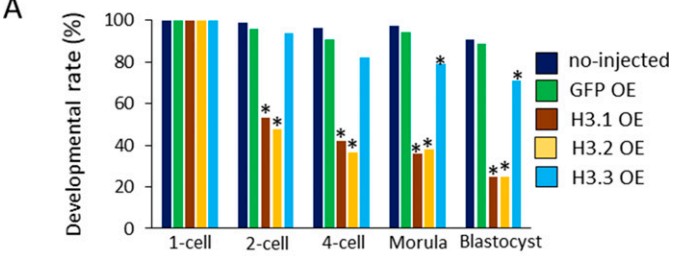

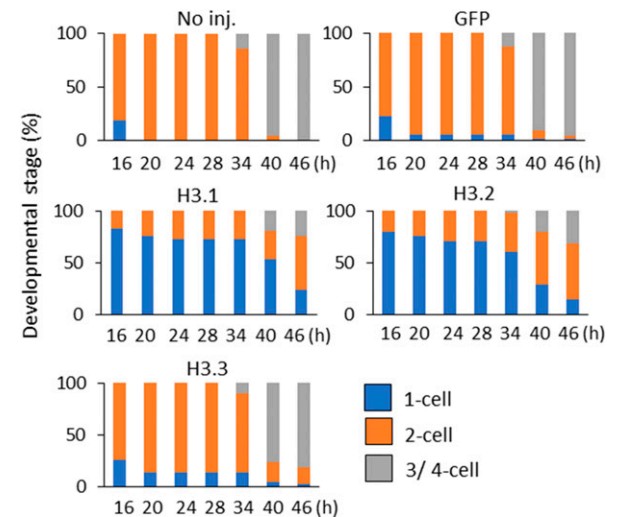

**Figure 3. Fluorescence images depicting the effects of H3.1/2 and H3.3 overexpression on H3 variant nuclear localization in one-cell embryos.**
Noninjected control, H3.1-, H3.2-, and H3.3-overexpressing one-cell embryos at 11 h post-insemination (hpi) were fixed and examined for changes in the nuclear localization of H3.1/2 and H3.3, using anti-H3.1/2 and anti-H3.3 antibodies, respectively. Eight independent experiments were performed and 39–58 total embryos were examined. Representative images are shown for each experiment. White arrowheads indicate the presence of H3.1/2 in the perinucleolar region of the paternal pronuclei. Scale bar, 10 $\mu$m.

chromatin and analyzed its effect on preimplantation development. Our results suggested that the incorporation efficiency of H3.1 and H3.2 into one-cell embryonic chromatin was low, compared to the incorporation efficiency of H3.3 (Fig 2B). However, when a high concentration of cRNA (100 ng/μl) was microinjected into MII-stage oocytes, similar levels of incorporation for all three H3 variants were observed. Exploiting this phenomenon, we microinjected 100 ng/μl of cRNA encoding FLAG-tagged H3 variants to force one-cell embryos to incorporate the H3 variants into chromatin. Less than 20% of the one-cell embryos that had been microinjected with H3.1 or H3.2 cRNA cleaved to the two-cell stage; in contrast, >90% of embryos injected with H3.3 cRNA, as well as control embryos (noninjected and GFP cRNA injected), progressed to the two-cell stage (Fig S3).

To exclude the possibility that the additional amino acids introduced with the FLAG-tag were detrimental with respect to embryonic development, we microinjected cRNA encoding H3 variants without the FLAG-tag. We first examined the nuclear localization of H3 variants in microinjected embryos by immunostaining with anti-H3.1/2 and anti-H3.3 antibodies (Fig 3). The results showed that the level of H3.1/2 nuclear deposition increased in both maternal and paternal pronuclei of H3.1- and H3.2-overexpressing embryos (H3.1/2-OE), which occurred in tandem with a reduction of H3.3 incorporation (Fig 3). The reduced H3.3 level was more pronounced in the male pronucleus than in the female pronucleus. Although H3.1/2 was only localized to the perinucleolar region of maternal pronuclei in noninjected embryos (Fig 1B), H3.1/2 was also deposited in the perinucleolar region of paternal pronuclei in H3.1/2-OEs (Fig 3). To reduce noise, the embryos were treated with Triton X-100 before fixation, which removed any free histones in the nucleoplasm. The structure of the nucleolar precursor body was disrupted after Triton X-100 treatment, but an aggregated perinucleolar structure could be identified (Fig S4). Notably, enhanced incorporation of H3.1/2 and reduced incorporation of H3.3 were detected in Triton X-100 treated H3.1/2-OEs, similarly to embryos that had not been treated with Triton X-100 before fixation (Fig 3). Furthermore, enhanced incorporation of H3.3 and reduced incorporation of H3.1/2 were observed

**Figure 4. Developmental delay and failure in H3.1/2-overexpressing one-cell embryos.**
**(A)** Developmental rates of noninjected, GFP-, H3.1-, H3.2-, and H3.3-overexpressing embryos (OEs) during the preimplantation stage. The noninjected, GFP-, H3.1-, H3.2-, and H3.3-OEs were incubated and analyzed at the following times: two-cell (28 h post-insemination [hpi]), four-cell (45–46 hpi), morula (72 hpi), and blastocyst (96 hpi). Eleven independent experiments were performed. For each group, 7–40 embryos were observed for each experiment; 197–228 embryos were observed in total. Asterisks represent statistical significance in the following analyses: for H3.1- and H3.2-OEs, a $\chi^2$ test or Fisher's exact test (when there was a group in which the value was below 5) was performed and the results were considered significant when $P < 0.01$ for noninjected, GFP-, and H3.3-OEs; for H3.3-OEs, a $\chi^2$ test or Fisher's exact test was performed and the results were considered significant when $P < 0.01$ for both noninjected and GFP-OEs. **(B)** The analysis of developmental stage of noninjected, GFP-, H3.1-, H3.2-, and H3.3-OEs from 16–46 hpi. The developmental rates of noninjected, GFP-, H3.1-, H3.2-, and H3.3-OE were observed at intervals of 4–6 h. Three independent experiments were performed. In each experimental group, 8–27 embryos were observed per experiment; 41–71 embryos were analyzed in total.

in H3.3-overexpressing embryos (H3.3-OEs; Fig S4). These results suggested that the detected histones are deposited in the chromatin, and that an alteration in the chromatin distribution of H3.1/2 and H3.3 occurs in H3.1/2-OEs.

Next, we investigated the impact of ectopic deposition of H3.1 and H3.2 at the one-cell stage of preimplantation development. Drastic developmental defects were observed in H3.1/2-OEs (Fig 4A), such that only 50% of embryos proceeded to the two-cell stage. The detrimental effects of H3.1 and H3.2 overexpression were more prominent in blastocysts: only ~30% of H3.1/2-OEs reached this stage. These results suggested that the limitation of H3.1/2 nuclear localization at the one-cell stage is essential for preimplantation development.

### Ectopic deposition of H3.1/.2 in pronuclei at the one-cell stage delays DNA replication

To gain mechanistic insight into the developmental failure of H3.1- and H3.2-OEs, the developmental rates of noninjected, H3.1-, H3.2-, H3.3-, and GFP-OEs were observed at intervals of 4–6 h (Fig 4B). Approximately 80% of the noninjected, GFP-, and H3.3-overexpressing one-cell embryos had cleaved into two-cell embryos at 20 hpi, and into four-cell embryos at 40 hpi. However, more than 70% of H3.1- and H3.2-OEs had not yet cleaved into two-cell embryos at 20 hpi, and most of them remained at the one-cell stage at 34 hpi. They initiated cleavage after

34 hpi and more than 70% of them developed to two-cell and later stages at 46 hpi. After that, some developed to the blastocyst stage, whereas other embryos eventually fragmented (Fig 4A). These results suggest that cellular cleavage from the one-cell stage to the two-cell stage is delayed in H3.1- and H3.2-OEs, which eventually leads to developmental failure.

The delay in cleavage from the one-cell stage to the two-cell stage in H3.1- and H3.2-OEs may be caused by a delay in DNA replication. To address this possibility, we examined the incorporation of BrdU into pronuclei at several time points post-insemination (Fig 5A). In the maternal pronuclei of noninjected, GFP-, and H3.3-OEs,

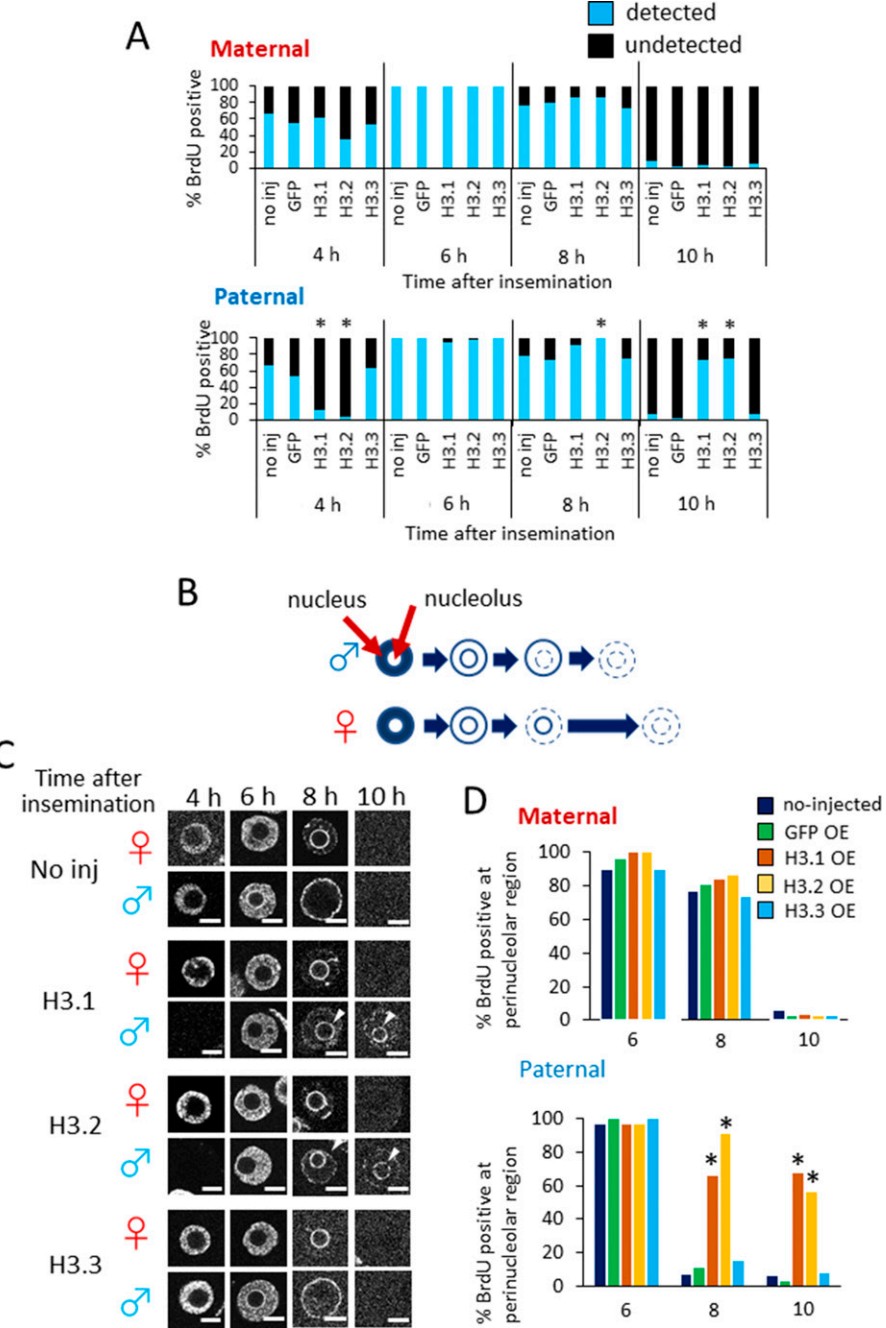

**Figure 5. Timing of DNA replication in noninjected, GFP-, H3.1-, H3.2-, and H3.3-overexpressing embryos.**
**(A)** The incorporation of BrdU was analyzed at 4, 6, 8, and 10 h post-insemination (hpi). Three to five independent experiments were performed. For each injected or noninjected sample, 30–51 embryos were analyzed in total. Asterisks represent statistical significance in the following analyses: for H3.1- and H3.2-overexpressing embryos (OEs), a $\chi^2$ test or Fisher's exact test was performed and the results were considered significant when $P < 0.01$ for noninjected, GFP-, and H3.3-OEs; for H3.3-OEs, a $\chi^2$ test or Fisher's exact test (when there was a group in which the value was below 5) was performed and the results were considered significant when $P < 0.01$ for noninjected and GFP-OEs. **(B)** Illustration of asymmetrical DNA replication in paternal and maternal pronuclei in one-cell embryos (Aoki & Schultz, 1999). DNA replication begins in the intranuclear region in both parental pronuclei. In the paternal pronucleus, DNA replication occurs first in the perinucleus, then in the perinucleolar region. In the maternal pronucleus, DNA replication in the perinucleolar region is completed after the paternal pronucleus has completed its replication. **(C)** Confocal images showing the patterns of BrdU incorporation in maternal (♀) and paternal (♂) pronuclei in noninjected, H3.1-, H3.2-, and H3.3-OEs at 4–10 hpi. For each injected or noninjected sample, 30–51 embryos were analyzed. Three to five independent experiments were performed. Representative images are shown for each experiment. Arrowheads indicate the presence of DNA replication at perinucleolar region in the paternal pronuclei of H3.1- and H3.2-OEs at 8–10 hpi. Scale bar, 10 μm. **(D)** Bar graph depicting %BrdU-positive perinucleolar regions for each microinjection condition at 6, 8, and 10 hpi. Three to four independent experiments were performed. For each sample, 19–51 embryos were analyzed in total. Asterisks represent the significant differences when compared with noninjected and GFP-OE embryos ($P < 0.01$, $\chi^2$ test).

DNA replication had initiated in >50% of embryos at 4 hpi. At 10 hpi, DNA replication was complete in most maternal pronuclei of non-injected, GFP-, and H3.3-OEs. Similarly, DNA replication in the maternal pronuclei of H3.1- and H3.2-OEs had initiated at 4 hpi and had been completed by 10 hpi. However, in H3.1- and H3.2-OEs, DNA replication was delayed in the paternal pronuclei; <15% of the paternal pronuclei had initiated DNA replication at 4 hpi, whereas only 25% had completed DNA replication at 10 hpi. These results suggested that the ectopic deposition of H3.1 and H3.2 into one-cell embryo chromatin leads to a delay in DNA replication in the paternal pronucleus, but not the maternal pronucleus.

## Ectopic deposition of H3.1 and H3.2 delays DNA replication in the perinucleolar region of paternal pronuclei

In somatic cells, DNA replication is completed earlier in euchromatic regions than in heterochromatic regions (O'Keefe et al, 1992). Previous reports have also shown that DNA replication occurs asynchronously between maternal and paternal pronuclei (Aoki & Schultz, 1999). In both paternal and maternal pronuclei, DNA replication begins in the nucleoplasm, then continues in the perinuclear and perinucleolar regions. In paternal pronuclei, DNA replication is completed first in the perinucleolar region, then in the perinuclear region; this contrasts with maternal pronuclear DNA replication, which is first completed in the perinuclear region, followed by the perinucleolar region (Fig 5B). Maternal pronuclear DNA replication requires additional time to complete, compared with paternal pronuclear DNA replication. Therefore, the time period required for perinucleolar replication in the female pronucleus determines the timing of S-phase completion.

As shown in Fig 3, the paternal pronuclei of H3.1 and H3.2-OEs displayed H3.1/2 nuclear distributions similar to those of maternal pronuclei. Considering these results, we hypothesized that the delay of DNA replication observed in the paternal pronuclei of H3.1 and H3.2-OEs (Fig 5A) was due to prolonged DNA replication in the perinucleolar region. To test this hypothesis, we observed the DNA replication sequences in maternal and paternal pronuclei at 4, 6, 8, and 10 hpi (Fig 5C). There were no significant differences in the sequence of DNA replication in the maternal pronuclei of noninjected, H3.1-, H3.2-, and H3.3-OEs; in all maternal nuclei, DNA replication occurred in the nucleoplasmic region at 4 and 6 hpi, continued in the perinucleolar region at 8 hpi, and was completed by 10 hpi.

However, the DNA replication pattern differed in the paternal pronuclei of H3.1- and H3.2-OEs; contrary to the noninjected control and H3.3-OEs, in which the perinuclear region was replicated last, DNA replication in the paternal perinucleolar region persisted at 10 hpi in H3.1- and H3.2-OEs (Fig 5C and D). We therefore concluded that the slowed cell cycle progression in H3.1- and H3.2-OEs was caused by a delay in DNA replication in the perinucleolar region of the paternal pronucleus. Furthermore, the initiation of DNA replication in the nucleoplasmic region was delayed for <2 hpi and was completed by 10 hpi, whereas DNA replication initiation in the perinucleolar region was delayed for >4 hpi and persisted at 10 hpi; these findings suggested that the delay of DNA replication in the perinucleolar region of the paternal pronucleus is the rate-limiting step, which delays cleavage in H3.1 and H3.2-OEs.

DNA replication in the maternal pronucleus of H3.1- and H3.2-OEs was unaffected by the induced incorporation of H3.1 and H3.2, which suggests that the maintenance of low H3.1/2 levels in the maternal pronucleus is not required for development. To confirm this suspicion, we generated parthenotes that were devoid of paternal genetic material and examined their developmental capacities when H3.1 and H3.2 had been introduced into their chromatin at the one-cell stage. As expected, there were no significant differences in developmental rate between H3.1- and H3.2-overexpressing parthenotes and H3.3-overexpressing parthenotes (Fig S5A). H3.1/2 incorporation was verified in H3.2-overexpressing parthenotes; the results indicated that the nuclear localization of H3.1/2 was similar to that of fertilized H3.2-OEs (Fig S5B). This strengthened the hypothesis that H3.1- and H3.2-OEs exhibit delayed cleavage due to H3.1 and H3.2 deposition in the paternal pronuclei, but not maternal pronuclei. Accordingly, it is essential that levels of H3.1/2 deposition is maintained at a low level in paternal pronuclei because enhancements of H3.1 and H3.2 deposition in the perinucleolar region of the paternal pronucleus can delay DNA replication, thereby leading to developmental failure.

## Effect of forced nuclear incorporation of H3.1 and H3.2 on epigenetic modifications

To further elucidate the molecular mechanisms underlying the delay in DNA replication in the paternal pronucleus of H3.1 and H3.2-OEs, we examined histone modification levels. The histone modifications H3K9me2/3 and H3K27me3 are involved in the formation of heterochromatin (Hake et al, 2006); they are often found on H3.1 and H3.2 in various cell types (Hake et al, 2006), and are unevenly detected in the parental pronuclei in one-cell-stage embryos (Lepikhov & Walter, 2004; Liu et al, 2004; Santos et al, 2005; Puschendorf et al, 2008).

Maternal pronuclear levels of the H3K9me2/3 modification are reportedly higher than the paternal levels (Liu et al, 2004; Lepikhov & Walter, 2004; Santos et al, 2005;; Puschendorf et al, 2008). Importantly, H3K9me3 was only detected in the maternal perinucleolar region, but not paternal perinucleolar region (Puschendorf et al, 2008). Given that the nuclear distribution of H3.1/2 in the paternal pronuclei became maternal pronucleus-like when H3.1 and H3.2 was overexpressed (Fig 3), we examined the methylation distribution on H3 variants to explore whether the methylation pattern in paternal pronuclei reflected the pattern in maternal pronuclei. However, the distributions of H3K9me2 and H3K9me3 modifications in the paternal pronuclei of H3.1- and H3.2-OEs did not differ from the control (Fig 6).

We then examined the distribution of the histone modification H3K27me3 in the H3 overexpression variants. For all overexpression conditions, the H3K27me3 signal was detected throughout the maternal pronucleus (except in the perinucleolar region) and presented a higher overall signal than the paternal pronucleus; however, in paternal pronuclei, H3K27me3 was clearly detected in the perinucleolar region (Fig 6). No differences in H3K27me3 nuclear distribution were observed between H3.1/2-OEs and control embryos.

We originally hypothesized that the H3K27me3 level would decrease in the paternal pronucleus of H3.1- and H3.2-OEs because K27 of H3.3 is methylated by PRC2 in the paternal perinucleolar region (Santenard et al, 2010; Tardat et al, 2015); we also observed a

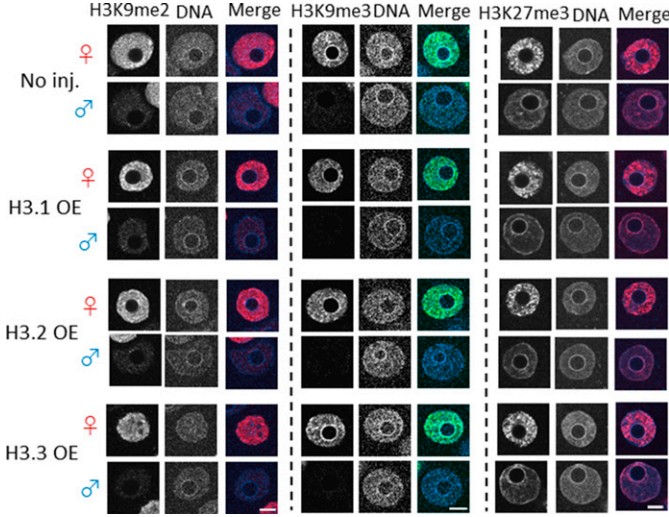

**Figure 6. Confocal images showing the effect of forced nuclear incorporation of H3.1, H3.2, and H3.3 on H3K9me2/3 and H3K27me3.**
Noninjected, H3.1-, H3.2-, and H3.3-overexpressing (OE) embryos were analyzed for methylation of H3K9me2, H3K9me3, and H3K27me3 at 11 h post-insemination. For H3K9me2 immunostaining, three independent experiments were performed, with the exception of H3.1-overexpressing embryos (two independent experiments). In total, 13–34 embryos were analyzed. For H3K9me3 immunostaining, two independent experiments were performed. 9–14 embryos were observed in total. For H3K27me3 immunostaining, four independent experiments were performed, with the exception of H3.1-overexpressing embryos (three independent experiments); in total, 23–32 embryos were analyzed in each experimental group. Representative images are shown for each experiment. Scale bar, 10 μm.

reduction in H3.3 levels in the paternal pronucleus in H3.1- and H3.2-OEs (Fig 3). However, no reduction in H3K27me3 levels was observed in the perinucleolar region of the paternal pronucleus in H3.1- and H3.2-OEs (Fig 6); this suggested that the H3.1/2 that displaced H3.3 in the paternal pronuclei of H3.1/2-OEs had acquired the K27me3 modification.

### Methylation of K27 of H3.1 and H3.2 in paternal perinucleolar chromatin causes developmental failure

Overexpression of H3.1 and H3.2 in one-cell embryos led to enhanced H3.1/2 and reduced H3.3 in the paternal perinucleolar region, whereas no reduction in H3K27me3 modification was observed (Fig 6). Therefore, we hypothesized that the delay in DNA replication could be caused by the ectopic methylation of H3.1 and H3.2 at the K27me3 residue in the paternal perinucleolar region of H3.1/2-OEs. The physiological significance of histone modifications has successfully been probed by microinjection of embryos that contain cRNA encoding H3 variants with amino acid substitutions (Santenard et al, 2010; Hatanaka et al, 2015; Zhou et al, 2017). To investigate the function of H3K27me3 in paternal pronuclei, we performed microinjection of cRNA encoding H3.1 and H3.2 with an arginine (R) substitution at residue 27 to replace K27 (H3.1K27R and H3.2K27R). The expression and incorporation of the mutant H3 proteins could not be verified directly using an anti-H3.1/2 antibody test, because K27 is part of the peptide sequence recognized by the antibody. H3.1/2 incorporation into pronuclei was instead verified indirectly through H3.3 displacement; H3.3 decreased in H3.1K27R-

and H3.2K27R-microinjected embryos in a manner similar to that observed in H3.1 and H3.2-OEs (Fig 7A). The H3K27me3 level in the perinucleolar region of paternal pronuclei decreased when H3.1K27R and H3.2K27R were overexpressed (Fig 7B).

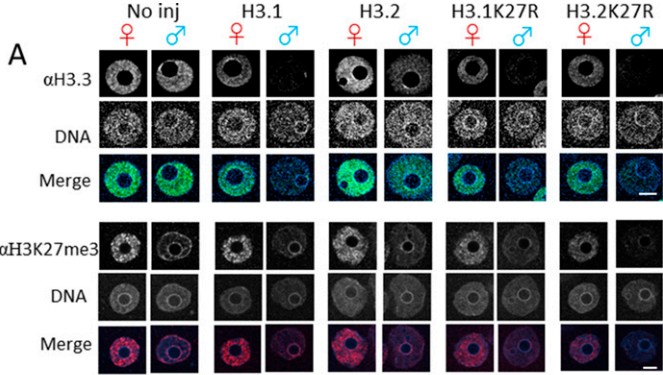

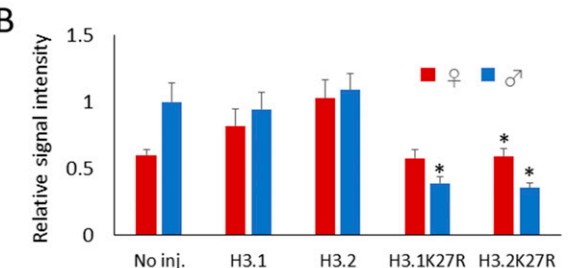

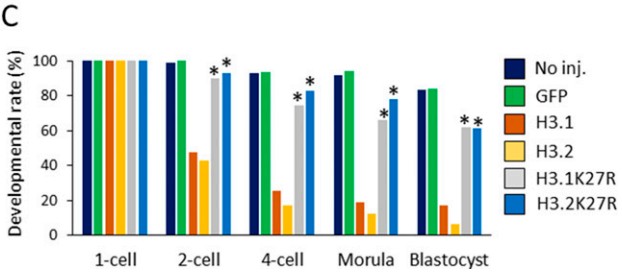

**Figure 7. Involvement of H3K27me3 modification on H3.1/2 in the developmental failure observed in H3.1- and H3.2-overexpressing embryos.**
**(A)** Confocal images of noninjected, H3.1/2-overexpressing, and H3.1/2K27R-overexpressing embryos stained with anti-H3.3 and anti-H3K27me3 antibodies. Three (αH3.3) or four (αH3K27me3) independent experiments were performed; 17–25 embryos (αH3.3) or 37–38 embryos (αH3K27me3) were analyzed in each experimental group. Representative images for each experiment are shown. Scale bar, 10 μm. **(B)** Bar chart showing the relative signal intensity for detected H3K27me3. H3K27me3 levels in the perinucleolar region were measured in noninjected, H3.1-, H3.2-, H3.1K27R-, and H3.2K27R-overexpressing embryos. Three independent experiments were performed; 27–31 embryos were analyzed in each experimental group. H3K27me3 signal measured at three perinucleolar sites and two background sites was used to quantify the signal intensity of H3K27me3 in the perinucleolar region for each pronucleus. The averaged signal intensity for H3K27me3 of male pronuclei in noninjected embryos was normalized to 1. Bars indicate standard error. Asterisks indicate significant differences in relative H3K27me3 signal intensity between H3.1K27R- or H3.2K27R-overexpressing embryos and H3.1- and H3.2-overexpressing embryos ($P < 0.01$, $t$ test). **(C)** Bar graph showing the developmental rates of noninjected, GFP-, H3.1-, H3.2-, H3.1K27R-, and H3.2K27R-overexpressing embryos. Five independent experiments were performed and 98–128 embryos were analyzed in total for each experimental group. Asterisks represent significant differences between developmental rates of H3.1K27R/H3.2K27R-overexpressing embryos and H3.1/H3.2-overexpressing embryos ($P < 0.01$, $\chi^2$ test).

We then examined the developmental rate of the H3.1/2 mutant overexpression lines (Fig 7C). Greater than 90% of the H3.1K27R and H3.2K27R-OEs progressed to the two-cell stage and >60% proceeded to the blastocyst stage, whereas <50% of H3.1- and H3.2-OEs developed to the two-cell stage and only 20% proceeded to the blastocyst stage. This strongly suggested that the H3K27me3 modification on H3.1/2 (H3.1/2K27me3) was the determining factor for the delay in DNA replication and subsequent developmental failure. Because K27 is also subjected to acetylation (K27ac) as well as methylation, it would be possible that the absence of K27ac could affect the development. However, this possibility can be excluded because K27ac is originally absent from perinucleolar region in one-cell stage embryos (Fig S6).

# Discussion

In this work, we have demonstrated that the deposition of histone variants in pericentromeric heterochromatin is asymmetrical between the paternal and maternal pronuclei. Moreover, the absence of H3.1 and H3.2 in the pericentromeric heterochromatin of the paternal pronucleus, but not maternal pronucleus, is essential for preimplantation development. The ectopic deposition of H3.1 and H3.2 in the paternal pericentromeric heterochromatin caused a delay in DNA replication, resulting in developmental failure. The detrimental effects of H3.1 and H3.2 paternal perinucleolar deposition on development were mitigated when the H3.1/2K27 residue was substituted for R, suggesting that trimethylation of K27 was responsible for the delay in DNA replication.

Epigenetic modifications have recently become the foci of intense academic study, bringing to light asymmetries of modifications between parental pronuclei (Hemberger et al, 2009; Burton & Torres-Padilla, 2010; Beaujean, 2014; Eckersley-Maslin et al, 2018). However, the biological significance of many such asymmetries has not yet been revealed. For example, the mechanisms regulating the parental asymmetry of global DNA methylation have been well researched, but biological roles for the asymmetries have not yet been established; however, some reports have suggested that these asymmetries are not involved in the regulation of development (Beaujean et al, 2004; Tsukada et al, 2015). In the present study, we propose that, rather than epigenetic modification alone, the combination of H3 variants and histone modifications (i.e., H3.1/2 with K27me3) determine the differences in DNA replication patterns between parental nuclei.

It was suggested in a previous study that DNA replication in the perinucleolar region of the maternal pronucleus is the rate-limiting step for cleavage from the one-cell stage to the two-cell stage (Aoki & Schultz, 1999). An investigation of last-replicating DNA regions in syncytial cycles of *Drosophila* embryos showed that extension of the S phase occurred as a result of delayed DNA replication in pericentric regions (Shermoen et al, 2010; Su, 2010). This finding suggested that replication of the pericentromeric region is the rate-limiting step for completion of the S phase; furthermore, prolonged replication of the pericentromeric heterochromatin in the paternal pronucleus might lead to delayed cleavage in H3.1 and H3.2-OEs.

It is unclear from our results whether the ectopic deposition of H3.1 and H3.2 led to developmental failure, or whether this failure

occurred following reduction in the nuclear deposition of H3.3 in the H3.1/2-OEs. In previous studies, H3.3 knockdown models caused developmental arrest (Lin et al, 2013); the deletion of HIRA, a chaperone of H3.3, caused a reduction of DNA replication in both parental pronuclei (Lin et al, 2014). However, the ectopic deposition of H3.1/2 presumably led to developmental failure, because DNA replication in the perinucleolar region was only delayed in the paternal pronucleus (Fig 5), which reflected the pattern of ectopic deposition of H3.1 and H3.2 in the same region (Fig 3). One report showed that the depletion of H3.3 in the paternal pronucleus prevented the incorporation of other core histones or histone variants (H2A and H2A.X) and led to abnormalities in the nuclear envelope and in nuclear transport (Inoue & Zhang, 2014); it is possible that DNA replication is impeded in abnormally formed pronuclei. However, our H3.1- and H3.2-OEs did not exhibit deformed or undersized pronuclei (Fig 3). Furthermore, Lin et al (2014) showed that the depletion of H3.3 triggered rRNA transcription, and that drug-induced inhibition of rRNA transcription caused cell cycle arrest at the one-cell stage; it did not cause inhibition of DNA replication.

The forced incorporation of H3.1 and H3.2 affected transcription at the one-cell stage. Transcriptional activity was assayed by measurement of embryonic BrU incorporation; this activity was significantly different between the parental pronuclei of H3.2-OE and control embryos, and different between the paternal pronuclei of H3.1-OE and no injected embryos. This activity was significantly different between the paternal pronuclei of H3.1- and H3.2-OEs and the control (Fig S7A). In contrast, no differences were detected between female pronuclei among injection conditions. The reduction in transcriptional activity could not have caused the delay in cleavage into the two-cell stage, as a previous study showed that zygotes treated with transcription-inhibiting α-amanitin cleaved to the two-cell stage normally (Warner & Versteegh, 1974). We initially anticipated that the rate of transcription of major satellite repeats would be altered by the forced incorporation of H3.1 and H3.2 in the male pronucleus because major satellite repeats are localized to the pronuclear rim (Probst et al, 2010) and are actively expressed in one-cell-stage embryos (Puschendorf et al, 2008; Probst et al, 2010). However, no significant changes in major satellite expression levels were detected by RT-PCR in H3.1- or H3.2-OEs (Fig S7B). Similarly, a recent study showed that ectopic expression of SUV39H1 in one-cell embryos to increase the level of H3K9me3, which generally suppresses transcription in a manner similar to that of H3K27me3, did not affect transcription; however, it had a detrimental effect on preimplantation development (Burton et al, 2020).

We propose the following model, in which the combination of correct H3 variants and heterochromatin-associated histone modifications is essential for the regulation of preimplantation development; alteration of this combination in the paternal perinucleolar region affected the timing of DNA replication, thus leading to developmental arrest (Fig 8). Immunocytochemical analyses showed that there were no differences in H3K9me2 or me3 levels between H3.1- and H3.2-overexpressing and noninjected control embryos (Fig 6). This finding is supported by a report that H3K9 methyltransferase is not functional in one-cell embryos, whereas there is methylation activity in the oocytes (Liu et al, 2004). Given that newly incorporated H3.1 and H3.2 could not be methylated

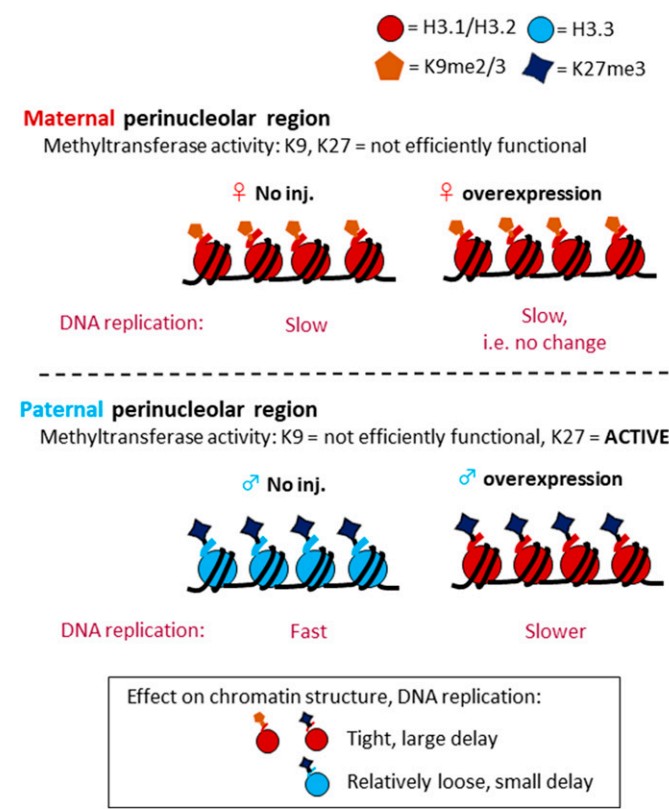

**Maternal perinucleolar region**
Methyltransferase activity: K9, K27 = not efficiently functional

♀ No inj.    ♀ overexpression

DNA replication:    Slow    Slow, i.e. no change

**Paternal perinucleolar region**
Methyltransferase activity: K9 = not efficiently functional, K27 = **ACTIVE**

♂ No inj.    ♂ overexpression

DNA replication:    Fast    Slower

Effect on chromatin structure, DNA replication:

Tight, large delay

Relatively loose, small delay

**Figure 8. Illustration depicting proposed mechanism for DNA replication delay in the paternal pronucleus of H3.1- or H3.2-overexpressing embryos.** In the perinucleolar region, K27 methyltransferase is active only in the paternal pronucleus. Ectopically incorporated H3.1/2 is trimethylated at the H3K27 in the paternal pronucleus alone. H3.1/2-H3K27me3 may contribute to the formation of a tight chromatin structure, leading to a delay in DNA replication in the perinucleolar region of the paternal pronucleus. H3.1/2 deposition in the pronuclei of zygotes is limited by reducing mRNA expression and histone incorporation into chromatin.

on K9 at the one-cell stage, the H3K9me2/3 levels were unaltered; the only H3K9me2/3 present had been carried over from the oocyte stage (Fig 8). Therefore, the level of K9me2/3-modified H3.1/2 (H3.1/2K9me2/3) is unchanged in all conditions, and there is no effect on the DNA replication timing in the maternal pronucleus (Fig 8). Similarly, the H3K27me3 level was not altered in the maternal pronuclei of H3.1- and H3.2-OEs (Fig 6) because PRC2 (a protein complex that exhibits H3K27 methyltransferase activity) is inhibited by the presence of HP1β (Burton et al, 2020). It has also been reported that heterochromatin protein 1β prevents PRC2 from binding in the maternal perinucleolar region (Tardat et al, 2015). However, PRC2 is functional in the perinucleolar region of the paternal pronucleus. In H3.1- and H3.2-OEs, PRC2 was able to methylate the newly incorporated H3.1 and H3.2 that had replaced H3.3 in the paternal perinucleolar region. Therefore, H3.1/2K27me3 increased in this region. A previous study showed that in one-cell embryos, the H3K27me3 modification was present on H3.3 in the paternal perinucleolar region (Santenard et al, 2010). H3K27me3 is associated with facultative heterochromatin, whereas H3.3 is mostly associated with euchromatin except for telomeres (Hake et al, 2006; Hake & Allis, 2006); we therefore hypothesize that H3.3K27me3 forms

heterochromatin with a loose structure, relative to H3.1/2K27me3 (Fig 8), although we cannot exclude other possibilities, for example, an effect of chromatin environment on the efficiency of replication origin firing in the repetitive regions and other histone modifications preferring H3.1/H3.2. Our hypothesis is supported by a report that DNA is replicated later in regions with higher levels of H3K27me3 modification in somatic cells (Thurman et al, 2007); in one-cell embryos, DNA in the paternal perinucleolar region (with H3.3K27me3) is replicated before the maternal perinucleolar region (with H3.1/H3.2K9me2/3) (Fig 5C). However, when the newly incorporated H3.1/2 is modified with K27me3 in the paternal perinucleolar region of H3.1- and H3.2-OEs, H3.1/2K27me3 may promote a tighter and more condensed perinucleolar region, compared with that of noninjected control embryos; this leads to a delay in DNA replication in that region. There are two possible mechanisms by which DNA replication could be delayed: first, the combination of H3.1/2 and K27me3 may have a greater effect on tightening of chromatin structure, compared to H3.1/2K9me2/3; second, the level of K27me3-modified H3.1/2 may be greater than that of K9me2/3-modified H3.1/2, due to the presence of PRC2 activity (Tardat et al, 2015) and the absence of K9 methylation activity in one-cell embryos (Liu et al, 2004).

The nuclear deposition of H3.1/2 is low at the one-cell stage, relative to the other preimplantation stages. We had originally hypothesized that limited nuclear deposition of H3.1/2 at the one-cell stage could be caused by low H3.1/2 expression and/or incorporation efficiency into chromatin. The mRNA levels of genes encoding H3.1 and H3.2 were lower at the one-cell stage than at other preimplantation stages (Fig 2A). Furthermore, H3.1/2 mRNA levels were lower than H3.3 mRNA levels at the one-cell stage. The efficiency of nuclear incorporation of H3.1 and H3.2 was also lower than that of H3.3 at the one-cell stage (Fig 2B). This low efficiency might have been caused by low expression of CAF1, which is a chaperone of H3.1/2. Our RT-PCR analysis revealed that the transcript level of a CAF1 component, *Caf1b*, was lower at the one-cell stage than at other preimplantation stages (data not shown). Although the expression levels of H3.1, H3.2, and *Caf1b* are low in one-cell embryos, their transcripts are present at detectable levels. At the one-cell stage, most mRNA transcripts are derived from oocytes, some of which are post-transcriptionally regulated (Yu et al, 2016; Sha et al, 2017); thus, H3.1, H3.2, and/or CAF1b proteins might be expressed at their lowest levels at this stage.

We propose that the localization of H3.1 and H3.2 is limited in one-cell embryos to prevent the detrimental effects elicited by the deposition of these proteins in the paternal pronucleus, thus preventing developmental failure. In the paternal pronucleus, the nuclear localization of H3.1/H3.2 to the perinucleolar region is equivalent to (or less than) localization to the perinuclear regions. Therefore, DNA replication in the perinucleolar region is completed before replication in the perinuclear region in the paternal pronucleus. In contrast, H3.1/H3.2 is localized to the perinucleolar region of the maternal pronucleus. Thus, DNA replication in this region is completed last in the maternal pronucleus. The enhanced deposition of H3.1/H3.2 caused the delay in completion of DNA replication in the paternal pronucleus, specifically in the perinucleolar region where pericentromeric heterochromatin is localized. Therefore, in one-cell embryos, this proposed mechanism

is required to reduce the deposition of H3.1/H3.2 in the paternal pronucleus by decreasing the overall mRNA expression and the efficiency of chromatin incorporation of H3.1 and H3.2.

# Materials and Methods

### Culture condition

All oocytes and embryos were incubated in droplets of medium covered in mineral oil (Sigma-Aldrich). The samples were incubated with 5% $CO_2$ at 38°C.

### Collection of preimplantation embryos

MII-stage oocytes were collected from 3-wk-old BDF1 (DBA2 × B6Ncr Jms Slc) mice (SLC Japan, Inc.; CLEA Inc.). Mice were injected with six I.U. pregnant mare's serum gonadotropin (ASKA Pharmaceutical Co, Ltd) followed by 7.5 I.U. of human chorionic gonadotropin (ASKA Pharmaceutical Co., Ltd) at 46–50 h after injection of pregnant mare's serum gonadotropin. Oviducts were removed from mice at 14–18 h after human chorionic gonadotropin injection. Mature oocytes surrounded by cumulus cells were collected and placed into 200 µl human tubal fluid medium (Quinn & Begley, 1984) supplemented with 10 mg/ml BSA (Sigma-Aldrich). In vitro fertilization was performed to obtain preimplantation embryos by insemination of oocytes with capacitated sperm, which had been preincubated for 2 h. At 3–7 hpi, the embryos and/or unfertilized oocytes were washed in K+-modified simplex optimized medium (KSOM) medium (Lawitts & Biggers, 1993). Pronuclei were examined at 6–10 hpi and cultured until they reached the blastocyst stage.

In vitro fertilization of denuded oocytes was conducted for microinjection analyses. Capacitated sperm were placed into 50 µl human tubal fluid medium (supplemented with BSA) and incubated for 1–2 min. The oocytes were then placed into the same medium. Embryos and/or unfertilized oocytes were washed in KSOM. Embryos at the one-cell, two-cell, four-cell, morula, and blastocyst stages were collected or observed at 10–11, 28–30, 45–46, 72, and 96 hpi, respectively.

All procedures using animals were reviewed and approved by the University of Tokyo Institutional Animal Care and Use Committee (#C-15-02) and were performed in accordance with the Guiding Principles for the Care and Use of Laboratory Animals.

### Immunocytochemistry

To detect nuclear localization of H3.1/H3.2 and H3.3, preimplantation embryos were fixed with 3.7% PFA and 0.2% Triton X-100 in PBS for 20 min at room temperature. Oocytes and preimplantation embryos were washed in PBS containing 1% BSA (BSA/PBS) and incubated overnight with mouse anti-H3.1/H3.2 (1:500; CE-039B; Cosmo Bio) or rat anti-H3.3 (1:100; CE-040B; Cosmo Bio) antibodies in BSA/PBS containing 0.2% Tween-20. The samples were then washed in BSA/PBS and incubated with Alexa Fluor 488 anti-mouse or rat IgG secondary antibodies (1:100; Molecular Probes, Invitrogen) for 1 h at room temperature. The samples were washed in BSA/PBS and mounted on a glass slide with Vectashield mounting media (Vector Laboratories) containing 1.6 ng/µl DAPI. For detection of histones that had been incorporated into chromatin, the procedures described by Hajkova et al (2010) were followed. FLAG-tagged histones were detected using anti-FLAG (1:1,000; Sigma-Aldrich) and Alexa Fluor 568 anti-rabbit IgG (1:100) antibodies. Endogenous histones were detected using anti-H3.1/H3.2 and anti-H3.3 antibodies with the dilutions described above.

For analysis of histone H3 dimethylated at lysine 9 (H3K9me2), trimethylated at lysine 27 (H3K27me3) or acetylated at lysine 27 (H3K27ac), the embryos were fixed in 3.7% PFA/PBS for 1 h or 20 min, respectively, and permeabilized with 0.5% Triton X-100 for 15 min. The mouse anti-H3K9me2 (ab1220; Abcam), anti-mouse H3K27me3 (05-851; Upstate/Millipore) and anti-mouse H3K27ac (C15410196; Diagenode) antibodies were diluted 1:100 in 1% BSA/PBS. For detection of H3K9me3, the embryos were fixed in 3.7% PFA/PBS containing 0.2% Triton X-100 for 20–25 min, then incubated with a rabbit anti-H3K9me3 antibody (04-772; Upstate/Millipore) that was diluted 1:100. For secondary antibodies, Alexa Fluor 568 anti-mouse IgG or Alexa Fluor 488 anti-mouse (Molecular Probes), or fluorescein isothiocyanate-conjugated donkey anti-rabbit (Jackson ImmunoResearch Inc.), were applied; slides were prepared as described above.

The samples were observed under a Carl Zeiss LSM5 exciter laser scanning confocal microscope (Carl Zeiss).

### Plasmid construction

The vector eGFP-polyA pcDNA3.1 (Yamagata et al, 2005) was used to generate Kozak-GFP cRNA as a control for microinjection. This vector was used as the backbone for other constructed vectors. The sequences of H3.1, H3.2, and H3.3 were described by Akiyama et al (2011). The sequences for H3.1K27R and H3.2K27R are as follows:

H3.1K27R: ATGGCTCGTACTAAGCAGACCGCTCGCAAGTCTACCGGCGGCA-AGGCCCCGCGCAAGCAGCTGGCCACCAAGGCCGCCCGCAGGAGCGCCCCGG-CCACCGGCGGCGTGAAGAAGCCTCACCGCTACCGTCCCGGCACTGTGGCGCT-GCGCGAGATCCGGCGCTACCAGAAGTCGACCGAGCTGCTGATCCGCAAGCTG-CCGTTCCAGCGCCTGGTGCGCGAGATCGCGCAGGACTTCAAGACCGACCTGC-GCTTCCAGAGCTCGGCCGTCATGGCTCTGCAGGAGGCCTGTGAGGCCTA-CCTCGTGGGTCTGTTTGAGGACACCAACCTGTGCGCCATCCACGCCAA-GCGTGTCACCATCATGCCCAAGGACATCCAGCTGGCCCGTCGCATCCGCG-GGGAGAGGGCTTAA

H3.2K27R: ATGGCTCGTACGAAGCAGACCGCTCGCAAGTCCACTGGCGG-CAAGGCCCCGCGCAAGCAGCTGGCCACCAAGGCCGCCCGCAGGAGCGCCCC-GGCCACCGGCGGCGTGAAGAAACCTCACCGCTACCGTCCCGGCACCGTGGCG-CTGCGCGAGATCCGGCGCTACCAGAAGTCGACCGAGCTGCTGATCCGCAAG-CTGCCGTTCCAGCGCCTGGTGCGCGAGATCGCGCAGGACTTCAAGACCGAC-CTGCGCTTCCAGAGCTCGGCCGTCATGGCTCTGCAGGAGGCGAGCGAGGCCT-ACCTTGTGGGTCTGTTTGAGGACACCAACCTGTGCGCCATCCACGCCAAGCG-TGTCACCATCATGCCCAAGGACATCCAGCTGGCCCGCCGTATCCGCGGCGAGCGG-GCTTAA

### cRNA microinjection

Plasmids were linearized and purified for cRNA generation. In vitro transcription was performed using T7 mMESSAGE mMACHINE kit (Ambion). Mature oocytes were collected in alpha-minimal essential

medium ($\alpha$-MEM) (Gibco-BRL) containing 5% FBS(Sigma-Aldrich) and 10 ng/ml EGF(Sigma-Aldrich). To remove the cumulus cells, hyaluronidase (Sigma-Aldrich) at a final concentration of 300 $\mu$g/ml was added to the medium and incubated for 5 min at 38°C and 5% $CO_2$. cRNA microinjection into mature oocytes was performed in Hepes-buffered KSOM, using an inverted microscope (Eclipse TE300; Nikon Corporation) with an attached micromanipulator and microinjector (Narishige Co.). cRNA was microinjected into the mature oocytes at a concentration and volume of 100 ng/$\mu$l and 10 pl, respectively. After microinjection, the oocytes were washed in $\alpha$-MEM (Gibco-BRL) containing 5% FBS and 10 ng/ml EGF. The oocytes were microinjected at 1.5–5 h after oocyte collection and incubated for another 2 h in $\alpha$-MEM to allow translation of injected histones before in vitro fertilization. The microinjected one-cell embryos were washed in KSOM medium, and then incubated at 38°C and 5% $CO_2$ until they reached the blastocyst stage. The method for in vitro fertilization of denuded oocytes is described above.

### Parthenogenesis

Parthenogenetic embryos were produced in accordance with the procedure described by Kishigami and Wakayama (2007). Mature oocytes were microinjected within 1.5–5 h after oocyte collection and incubated for another 2 h to allow translation of histones in $\alpha$-MEM containing 5% FBS and 10 ng/ml EGF. The mature oocytes were then activated by incubation for 3 h (38°C with 5% $CO_2$) in KSOM containing 2 mM EGTA, 5 mM SrCl$_2$, and 5 $\mu$g/ml Cytochalasin B (Sigma-Aldrich) to generate parthenogenetic embryos with two pronuclei. Parthenogenetic embryos with two pronuclei were produced to generate embryos with the same numbers of histones as the in vitro fertilized embryos. After 3 h of activation, the embryos were washed with KSOM; parthenogenetic embryos with two pronuclei were selected and cultured in KSOM until the blastocyst stage.

### mRNA expression analysis of H3 variants

The RPKM values of genes encoding H3.1 (*Hist1h3a*, *Hist1h3g*, *Hist1h3h*, *Hist1h3i*), H3.2 (*Hist1h3f*, *Hist1h3b*, *Hist1h3d*, *Hist1h3e*, *Hist2h3b*, *Hist1h3c*, *Hist2h3c2*, and *Hist2h3c1*), and H3.3 (*H3f3a* and *H3f3b*) were obtained from a previously published RNA-seq dataset (Abe et al, 2015). RPKM values for each H3 variant were totaled to compare total expression levels among the variants. The expression level of H3.3 at the one-cell stage was normalized to 1 and the relative expression levels of H3.1, H3.2 for one-cell to blastocyst stages, and H3.3 for two-cell to blastocyst were calculated; expression ratios were calculated for H3.1 and H3.2.

### Immunofluorescence quantification

The signal intensities of FLAG and H3K27me3 antibodies, as well as DAPI, were quantified using ImageJ software (National Institutes of Health). The signal intensities of FLAG of the pronuclei were subtracted by the average of two background areas in the cytoplasm, then corrected using the DAPI signal. The signal intensities of H3K27me3 at the perinucleolar region were calculated by quantifying three areas of perinucleolar regions (in which the average signal of two background areas in the cytoplasm was subtracted) and then corrected using the DAPI signal of these regions. The maternal and paternal pronuclei were identified by their size and proximity to the second polar body; the maternal pronucleus is smaller and proximal to the polar body.

### BrdU incorporation assay for DNA replication

DNA replication was analyzed by examination of BrdU incorporation in one-cell embryos at 4, 6, 8, and 10 hpi. BrdU (Roche) was added to the KSOM to a final concentration of 10 $\mu$M. After incubation at 38°C for 30 min, the embryos were washed three times in 1% BSA/PBS and fixed with 3.7% PFA/PBS for 1 h at room temperature. After fixation, the samples were washed with 1% BSA/PBS three times, then washed three times in PBS containing 0.05% Tween-20. The samples were then placed under 2 N HCl containing 0.1% Triton X-100 for 1 h at 37°C. Next, the samples were washed with 1% BSA/PBS three times and transferred into 0.1 M Tris–HCl (pH 8.5)/PBS containing 0.02% Triton X-100 for 15 min at room temperature. The samples were washed three times with 1% BSA/PBS and incubated overnight with primary antibodies: mouse anti-BrdU (1:100; Roche) and rabbit anti-H3K9me3 (1:1,000; Upstate/Millipore). Alexa Fluor 647 anti-mouse IgG (Molecular Probes) and fluorescein isothiocyanate–conjugated donkey anti-rabbit (Jackson Immuno-Research Inc.) were used as secondary antibodies; slides were prepared as described above.

### In vitro transcription assay

Total transcriptional activity was evaluated by the incorporation of BrUTP into nascent mRNA, as described previously (Kim et al, 2002). BrU signal intensity was quantified using ImageJ software (National Institutes of Health), following the same method as FLAG quantification.

### RT-PCR

H3.1-, H3.2-, H3.3-, and GFP-overexpressing embryos and noninjected embryos were placed in ISOGEN RNA extraction reagent (Nippon Gene); RNA extraction was conducted in accordance with the manufacturer's instructions. The extracted RNA was reverse transcribed using the PrimeScript RT-PCR Kit (TaKaRa). Real-time PCR was conducted as described by Kawamura et al (2012). Primers used to detect major satellite repeats were as described by Inoue et al (2012).

### Inhibition of DNA replication

DNA replication was inhibited by transferring one-cell embryos to KSOM supplemented with 3 $\mu$g/ml aphidicolin (Sigma-Aldrich) at 15 hpi and collected for sampling at 26 hpi. DMSO was used as the solvent for aphidicolin suspension; control embryos were cultured with DMSO.

# Supplementary Information

# Acknowledgements

We thank Tetsuya Kojima and Shoji Oda for their helpful discussions. This work was supported in part by Grants-in-Aid (to F Aoki) from the Ministry of Education, Culture, Sports, Science and Technology of Japan (#20062002, #25252054, 16H01215, 18H03970, and 19H05752).

## Author Contributions

M Kawamura: conceptualization, data curation, formal analysis, investigation, and visualization, methodology.
S Funaya: data curation, formal analysis, investigation, and methodology.
K Sugie: data curation, formal analysis, and methodology.
MG Suzuki: data curation and investigation.
F Aoki: conceptualization, data curation, formal analysis, funding acquisition, validation, visualization, and project administration.

## Conflict of Interest Statement

We declare that there is no conflict of interest that could be perceived as prejudicing the impartiality of the research reported.

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
