## [Reviewer comments · Life Science Alliance]

Life Science Alliance

Asymmetrical deposition and modification of histone H3 variants is essential for zygote development

Machika Kawamura, Satoshi Funaya, Kenta Sugie, Masataka Suzuki, and Fugaku Aoki

DOI: <https://doi.org/10.26508/lsa.202101102>

Corresponding author(s): Fugaku Aoki, University of Tokyo

Review Timeline:

Submission Date:	2021-04-21
Editorial Decision:	2021-05-26
Revision Received:	2021-05-27
Editorial Decision:	2021-06-08
Revision Received:	2021-06-14
Accepted:	2021-06-15

Transaction Report:

Please note that the manuscript was previously reviewed at another journal and the reports were taken into account in the decision-making process at Life Science Alliance.

Reviewer #1 Review

Comments to the Authors (Required):

Kawamura et al.

Assymetrical deposition and modification of histone H3 variants is essential for zygote development. This study examines the impact of canonical H3 and its variant H3.3 on the replication of male and female pronuclei in the one cell embryo in the hours after fertilisation. The authors build on previous work demonstrating asymmetry in the replication of the two pronuclei and in the deposition of the canonical, replication-dependent H3s (H3.1 and H3.2). The authors show that deposition of H3.1/2 is low in the first replication of the pronuclei due to a combination of low expression and inefficient incorporation. Overexpressing H3.1/2, but not H3.3 results in a delay in replication and developmental failure. The authors present a series of experiments, which I think are persuasive,

that this delay is caused by trimethylation of this deposited H3.1/2 at K27.

This is a very well executed, controlled and described study that I believe contains observations of significant interest. I do not have major criticisms of the data that is shown or its analysis. However, I think it is fair to say that the mechanism by which increase H3K27me3 in the paternal perinucleolar region delays replication remains unclear. While greater compaction is one possibility, there are a number of other potential explanations, for instance an effect of the chromatin environment on the efficiency of replication origin firing in this repetitive region. There is evidence linking H3K27me3 to replication origins (e.g. Picard et al PLoS Gen 2014), but whether it is really a regulator (indeed positively or negatively) remains rather unclear. Some additional discussion of this point might be warranted. Further, the reasons why H3.3 cannot substitute as a target for excessive H3K27 methylation is unclear. However, I do not believe that establishing these points is for this manuscript.

I was concerned when reading the results that overexpression of the canonical H3.1/2 could lead to replication defects, as previously shown in somatic cells (e.g. Groth et al. Science 2007) and that this could contribute to the delay in replication. However, while the authors note in the discussion that they observe no increase in gammaH2Ax in the over-expressing cells, I think this is an important point and it would be good to show the data.

Minor points:

- The second sentence of the abstract may cause some confusion as it can be read to imply that the paper demonstrates asymmetric distribution between H3.1 and H3.2, rather than of them both together. Just needs slight rewording to make it clearer that these are grouped together.
- Figure 2B - X-axis label spelling error 'Fag' should be 'Flag'.
- The points in figure 2B should be joined by straight lines, not by the artificial curves generated in Excel.
- Statistics appear to be missing from some graphs e.g. 5D, even though the number of experiments and embryos is given in the legend.
- Page 21, line 8: stray full stop

Reviewer #2 Review

Comments to the Authors (Required):

On fertilisation epigenetic reprogramming re-establishes totipotency. In the course of this remodelling parental asymmetries are essential to support onward development. In this study the authors demonstrate that this asymmetry is maintained at the level of the histone H3 variants which comprise the chromatin of the respective pronuclei and specifically prevents ectopic introduction of H3.1 K27me3 in the paternal compartment. This careful dissection of the H3 variants, in the first cell cycle, highlight the importance of the fidelity of heterochromatin organisation in each parental compartment to establish the trajectory for further preimplantation development.

Comments to the Authors

In Fig 1 b we see pronuclei with opposing pattern of staining. Here it would be useful to consider another mark exclusive to each to reinforce their parental identity. In lieu of this, imaging to capture the entire fertilised oocytes with high mag blow out of the individual pronuclei to illustrate this differential pattern would be more informative than the DAPI staining and merged panel.

Pg7 line 9 - There are better primary data references to pericentromeric heterochromatin being to appear in chromocenters than the review Probst and Almouzni 2011. In Fig 1C the arrows identify the pericentric heterochromatin which occupies the pronuclear periphery but I do not see these in the DAPI channel, which has a uniform punctate pattern across the entire image making resolution of nuclear details impossible.

The authors suggest that the chromatin structure changes from H3.1K27me3 to H3.3 K27me3. The chromatin structure of pronuclei is uniquely open, hence the size of the pronuclei. Is this 'loose' chromatin structure consistent with larger pronuclei?

In many of the figures, the histone detection signal and the DNA channel have punctate signal both inside and outside of the pronuclei. It is not clear whether this is owing to very low signal and hence noise or another systematic artefact. In any case, it diminished the data's visual impact and should be cleaned up.

The authors have extracted expression data from previous data from their work, however, RNA-seq data on specific transcripts does not convert easily into data where statistical analyses can be done. As such, how was this data evaluated?

Discussion indicates that DNA damage and H2AXg were not increased on OE. These data should be included in the supplemental data section for completeness.

Minor comments

1. The figure legends are not very informative. Inclusion of the details would be a welcome addition for the reader.
2. Nucleoli are not functional until the 8-cell stage in mouse embryos. Until this time these structures are referred to as nucleolar precursor bodies.
3. Pg 19 Line 2. Capitalisation missing.
4. Pg 18 reference Frey et al., should not include Current Biology.
5. Pg 21 Line 8 has a 'fullstop' erroneously at the end of the line.

Reviewer #3 Review

Comments to the Authors (Required):
Summary:

In this manuscript, Kawamura et al. investigate the nuclear patterns of histone H3.1/2 and H3.3 variants in preimplantation embryo development, more specifically 1-cell stage embryos.

In agreement with previous studies, the authors find that in murine zygotes, H3.1 and H3.2 are asymmetrically distributed between maternal and paternal pronuclei at the pericentromeric chromatin (PCH), with an relative enrichment on maternal pronuclei.

In brief, the new observations here are that when the authors overexpress H3.1/2 in 1-cell stage embryos, the variants can accumulate at paternal PCH. This is accompanied by a DNA replication delay of the PCH, and is further associated with a cleavage delay starting from the 2-cell stage. However, the abnormal accumulation of H3.1/2 in paternal PCH did not give rise to detectable changes in H3K9me2/3 or H3K27me3. Notably, previous work had shown that H3.3 K27 trimethylation at the paternal PHC was important for normal development (Santenard et al., Nat Cell Biol 2010). Thus, the authors overexpressed H3.1/2K27R in one-cell stage embryos, which surprisingly rescued the developmental delay present upon H3.1/2 overexpression. Based on these result they suggest that the K27 residue of H3.1/2 is important for mediating the effects observed upon H3.1/2 overexpression.

The manuscript does address an interesting issue of importance for developmental biology and epigenetics. However, at this stage the observations are not sufficiently solid and sometimes apparently inconsistent. The interpretation of the results is unclear and premature, since the authors do not provide direct evidence to support the fact that H3.1/2K27me3 is responsible for the PCH replication delay and developmental defects. There is also a need for them to revisit the background literature on the topic.

In conclusion, the proposed manuscript needs significant work to be considered for publication both experimentally and in the analysis to increase the quality of the data. The conclusions are rather speculative and the implications unclear. Some potential issues should also be addressed more carefully, and the authors should provide quantification of the results (since many conclusions are only supported by representative images). A list of suggestions is detailed below.

Major points:

- Throughout the study the authors state that they have measured chromatin incorporation of H3 variants. However, they have performed IF for total H3.1/2, H3.3 or H3.1/2/3-FLAG (with the exception of Suppl. Fig. S4), which does not allow them to distinguish between chromatin-bound and free histones (Fig. 1, 2B, 3, 6, 7 and Suppl. Fig. S1, S2, S5B). Hence, the authors effectively describe nuclear patterns of H3.1/2 or H3.3, but not their incorporation into chromatin, and this should be stated more clearly in the text. Most importantly, the evidence provided does not allow to infer that H3.1/2 incorporation at the paternal PCH is responsible for the delay.
- Similarly, the role of the K27 residue upon H3.1/2 overexpression in mediating DNA replication delay is unclear. Additional experiments directly demonstrating that incorporation of H3.1/2K27R in overexpressing embryos can prevent the replication delay should be performed. Furthermore, this mutation ablates not only methylation, but also acetylation of K27 (and may also impact neighbouring PTMs). This point should be stated more clearly. This is especially critical since the authors state their results 'strongly suggested that the H3K27me3 modification on H3.1/2 (H3.1/2K27me3) was the determining factor for the delay in DNA replication and subsequent developmental failure' (page 16, lines 13-15). This is an important conclusion if true, but to make a strong argument this should be supported by further experiments showing that the effect is directly mediated by K27 trimethylation.
- Many of the conclusions brought forward by the authors are only supported by representative IF images (Figs. 1, 3, 6 and Suppl. Figs. S2, S4 and S5) without quantifications of the signal or observed patterns at the maternal/paternal PHC or summary statistics. The authors should provide quantification and statistics to better corroborate their findings. Furthermore, error bars representing the standard deviation between experiments should be included to show variability in

all bar plots (Fig. 4, 7C, Suppl. Fig S3, S6A) of IF signal (Fig. 5A, D).

- The authors state 'maternal and paternal perinucleolar regions were enriched in H3.1/2 with K9me2/3 and H3.3 with K27me3, respectively' (page 5, lines 17-18). However, in the images shown here, no clear enrichment of the variant is detectable at the paternal PHC (Fig. 1A, anti-H3.3, Fig. 3, neither no injection, nor H3.3 OE, Fig. 7A, Suppl. Fig. S4), although this has been previously described in the literature (Santenard et al., Nat Cell Biol 2010, Fig. 5a, b, Liu et al., EMBO J 2020, Fig. 1A PN5). How do the authors explain this discrepancy?
- Suppl. Fig. S6 is not described in the Results and is only introduced in the Discussion. It is not clear what the purpose of the data shown there is and how it fits with the rest of the manuscript.
- The model in Fig. 8 and many conclusions are largely speculative and not directly supported by the evidence presented in this manuscript. The authors did not investigate changes of neither chromatin structure nor the activity of lysine methyltransferases and they did not provide any data informing on their link to DNA replication timing. While some speculation can be stimulating here it seems that the basis for speculation is too weak, the model and the discussion of the data presented should be revised to show how it actually links to the authors' findings.

Other points:

- More details about the imaging system should be provided in the Methods section
- In Fig. 1C, nucleoli and chromocenters are not clearly distinguishable in the DAPI panel
- In Fig. 2B, the y-axis label of the plot should be fixed. Also, are the p-values reported in the legend corrected for multiple testing? The anti-FLAG staining (top panel) also does not show perinucleolar enrichment, this is in contrast with other images taken at the same stage with the anti-H3.1/2 or anti-H3.3 antibodies, why? How do the authors interpret this? Adding a DAPI panel for reference would also be helpful to display the chromatin density in these cells.
- In several of the figures (Fig. 4A, 5A and Suppl. Fig. S3, S5A) the authors state they performed 'chi-squared or Fisher's exact test'. The authors should better clarify which test they performed (or why they performed two tests, if this is the case)
- The authors state that the 'distributions of H3K9me2 and H3K9me3 modifications in the paternal pronuclei of H3.1- and H3.2-OEs did not differ from the control' (page 14, lines 19-20). However, there is an apparent reduction of H3K9me3 in all overexpressing embryos compared to no injection at both maternal and paternal pronuclei, and it appears to be less enriched at maternal PHC specifically in H3.1- and H3.2-OEs (Fig. 6).
- In Suppl. Fig. S2A, no H3.1/2 signal is detected at 15hpi embryo, whereas H3.1/2 is already present in the 11hpi Aphi (-) embryos in Suppl. Fig. S2B. How do the authors explain this?
- The number of embryos injected per experiment should be specified in Suppl. Fig. S4
- The statement that 'H3.1 is enriched in the nucleolus during the latter stages of chromatin replication, whereas H3.3 localizes during early-stage chromatin replication (Clément et al., 2018); this suggests that H3 variants play a role in the regulation of DNA replication' (page 11, lines 5-8) should be rephrased to accurately reflect the findings of the study cited, which shows the H3 variant distribution correlates with replication timing but doesn't make claims as to a causal impact.
- The statement that 'initiation of DNA replication in the nucleoplasmic regions was delayed for <2hpi and was completed by 10hpi' (page 13, lines 4-6) is not supported by the evidence presented, and should be rephrased.
- Similarly, the sentence 'PRC2 ... is not functional in the maternal perinucleolar region' (page 20, lines 8-10) is inaccurate: PRC2 is functional, but inhibited by the presence of HP1b at this region (Burton et al., Nat Cell Biol, 2020)
- The authors state page 20, line 18, 'H3K27me3 is associated with facultative heterochromatin, H3.3 is associated with euchromatin (Hake et al., 2006, Hake and Allis, 2006); we therefore hypothesize that H3.3K27me3 forms heterochromatin with a loose structure, relative to H3.1/2K27me3'. However, H3.3 has also been shown to be present at pericentromeric and telomeric

chromatin in embryonic stem cells and MEFs (Goldberg et al., Cell 2010 and Drané et al., Gen & Dev 2010, respectively). This shows an example of the need for the authors to re-examine the background literature.

- The authors discuss the incorporation efficiency of H3.1/2 at 1-cell stage embryos (page 21, lines 17-20) and link it to the expression levels of the variants (Fig. 2) and of Caf1b (one of the homologues of one of the subunits of the CAF-1 complex which deposits H3.1/2 on chromatin) which they measure by RT-PCR but do not show data for. The data supporting this statement should be included.

May 26, 2021

Re: Life Science Alliance manuscript #LSA-2021-01102-T

Dr. Fugaku Aoki
University of Tokyo
University of Tokyo
Kashiwanoha 5-1-5
Kashiwa 277-8562
Japan

Dear Dr. Aoki,

Thank you for submitting your manuscript entitled "Asymmetrical deposition and modification of histone H3 variants is essential for zygote development" to Life Science Alliance (LSA).

For a brief overview, this manuscript was previously reviewed at our partner journal, and the authors chose to transfer the manuscript and the reviewers' comments to LSA. LSA is willing to consider a revised version of the manuscript that addresses,

- + concerns raised by Reviewers 1 and 2
- + Reviewer 3's major concerns 1, 3 and 4 and all minor concerns
- + The other major concerns raised by Reviewer 3 can be addressed with discussion and toning down the conclusions

Please send me a point-by-point response to indicate the reviewers' comments that have been addressed with additional data and the comments that have been addressed with clarifications. We would also encourage you to share a marked up manuscript file with us that highlights the changes.

Please note that such a revision might require re-review, in which case, we will walk the reviewers through the transfer process.

Thank you for this interesting contribution to Life Science Alliance. We are looking forward to receiving your revised manuscript.

Sincerely,

Shachi Bhatt, Ph.D.
Executive Editor
Life Science Alliance
<http://www.lsajournal.org>
Tweet @SciBhatt @LSAJournal

- A letter addressing the reviewers' comments point by point.
- An editable version of the final text (.DOC or .DOCX) is needed for copyediting (no PDFs).
- High-resolution figure, supplementary figure and video files uploaded as individual files: See our detailed guidelines for preparing your production-ready images, <https://www.life-science-alliance.org/authors>
- Summary blurb (enter in submission system): A short text summarizing in a single sentence the study (max. 200 characters including spaces). This text is used in conjunction with the titles of papers, hence should be informative and complementary to the title and running title. It should describe the context and significance of the findings for a general readership; it should be written in the present tense and refer to the work in the third person. Author names should not be mentioned.

B. MANUSCRIPT ORGANIZATION AND FORMATTING:

Reviewer #1 (Comments to the Authors (Required)):

Kawamura et al.

Assymetrical deposition and modification of histone H3 variants is essential for zygote development

This study examines the impact of canonical H3 and its variant H3.3 on the replication of male and female pronuclei in the one cell embryo in the hours after fertilisation. The authors build on previous work demonstrating asymmetry in the replication of the two pronuclei and in the deposition of the canonical, replication-dependent H3s (H3.1 and H3.2). The authors show that deposition of H3.1/2 is low in the first replication of the pronuclei due to a combination of low expression and inefficient incorporation. Overexpressing H3.1/2, but not H3.3 results in a delay in replication and developmental failure. The authors present a series of experiments, which I think are persuasive, that this delay is caused by trimethylation of this deposited H3.1/2 at K27.

This is a very well executed, controlled and described study that I believe contains observations of significant interest. I do not have major criticisms of the data that is shown or its analysis. However, I think it is fair to say that the mechanism by which increase H3K27me3 in the paternal perinucleolar region delays replication remains unclear. While greater compaction is one possibility, there are a number of other potential explanations, for instance an effect of the chromatin environment on the efficiency of replication origin firing in this repetitive region. There is evidence linking H3K27me3 to replication origins (e.g. Picard et al PLoS Gen 2014), but whether it is really a regulator (indeed positively or negatively) remains rather unclear. Some additional discussion of this point might be warranted. Further, the reasons why H3.3 cannot substitute as a target for excessive H3K27 methylation is unclear. However, I do not believe that establishing these points is for this manuscript.

Thank you for a helpful suggestion. We have discussed about other potential explanations for the delay in DNA replication in H3.1/3.2OE embryos (page 20, line 15-18 in the revised manuscript).

I was concerned when reading the results that overexpression of the canonical H3.1/2 could lead to replication defects, as previously shown in somatic cells (e.g. Groth et al. Science 2007) and that this could contribute to the delay in replication. However, while

the authors note in the discussion that they observe no increase in gammaH2Ax in the over-expressing cells, I think this is an important point and it would be good to show the data.

We have reconsidered this point and found that the discussion about gammaH2A.X is not appropriate, because the level of gammaH2A.X is extremely low even in the irradiated (DNA damaged) embryos at the 1-cell stage (Yukawa et al., BBRC 358:578-584, 2007). Therefore, we have deleted the description relating to gammaH2A.X.

Minor points:

- The second sentence of the abstract may cause some confusion as it can be read to imply that the paper demonstrates asymmetric distribution between H3.1 and H3.2, rather than of them both together. Just needs slight rewording to make it clearer that these are grouped together.

The second sentence in the Abstract has been corrected to avoid the confusion.

- Figure 2B - X-axis label spelling error 'Fag' should be 'Flag'.

The spelling error has been corrected in Fig. 2B.

- The points in figure 2B should be joined by straight lines, not by the artificial curves generated in Excel.

The artificial curves have been changed to straight lines in Fig. 2B.

- Statistics appear to be missing from some graphs e.g. 5D, even though the number of experiments and embryos is given in the legend.

The results of statistical analysis are shown in Fig. 5D and the explanation for this analysis is described in the legend for Fig. 5D.

- Page 21, line 8: stray full stop

The period has been removed.

Reviewer #2 (Comments to the Authors (Required)):

On fertilisation epigenetic reprogramming re-establishes totipotency. In the course of this remodelling parental asymmetries are essential to support onward development. In this

study the authors demonstrate that this asymmetry is maintained at the level of the histone H3 variants which comprise the chromatin of the respective pronuclei and specifically prevents ectopic introduction of H3.1 K27me3 in the paternal compartment. This careful dissection of the H3 variants, in the first cell cycle, highlight the importance of the fidelity of heterochromatin organisation in each parental compartment to establish the trajectory for further preimplantation development.

Comments to the Authors

In Fig 1 b we see pronuclei with opposing pattern of staining. Here it would be useful to consider another mark exclusive to each to reinforce their parental identity. In lieu of this, imaging to capture the entire fertilised oocytes with high mag blow out of the individual pronuclei to illustrate this differential pattern would be more informative than the DAPI staining and merged panel.

According to the reviewer's suggestion, we have remade Fig. 1B to show the parental identity clearly: the images of double stain with H3K9me3 which is present only in female pronuclei.

Pg7 line 9 - There are better primary data references to pericentromeric heterochromatin being to appear in chromocenters than the review Probst and Almouzni 2011.

We have replaced the reference literature from appropriate one describing chromocenters appearing at the 2-cell stage (page 7, line 10 in the revised manuscript).

In Fig 1C the arrows identify the pericentric heterochromatin which occupies the pronuclear periphery but I do not see these in the DAPI channel, which has a uniform punctate pattern across the entire image making resolution of nuclear details impossible.

We have replaced the image in Fig. 1C to clearly show the DAPI signal and the merged localization of the signals of DAPI and H3.1/3.2.

The authors suggest that the chromatin structure changes from H3.1K27me3 to H3.3 K27me3. The chromatin structure of pronuclei is uniquely open, hence the size of the pronuclei. Is this 'loose' chromatin structure consistent with larger pronuclei?

We suggest the chromatin structure changes in centromeric heterochromatin of the "nucleolar peripheral regions." Therefore, the size of the "whole pronuclei" would not be affected by the changes from H3.1K27me3 to H3.3 K27me3.

In many of the figures, the histone detection signal and the DNA channel have punctate signal both inside and outside of the pronuclei. It is not clear whether this is owing to very low signal and hence noise or another systematic artefact. In any case, it diminished the data's visual impact and should be cleaned up.

We have conducted immunocytochemistry with the H3.1/3.2 antibody over again to obtain fine images in which noises is reduced and the image of Fig. 1B and 1C has been replace by new ones in the revised manuscript.

The authors have extracted expression data from previous data from their work, however, RNA-seq data on specific transcripts does not convert easily into data where statistical analyses can be done. As such, how was this data evaluated?

Since in the previous experiments of RNAseq, only duplicate samples were analyzed, it is inappropriate to conduct statistical analysis for them.

Discussion indicates that DNA damage and H2AXg were not increased on OE. These data should be included in the supplemental data section for completeness.

We have reconsidered this point and found that the discussion about gammaH2A.X is not appropriate, because the level of gammaH2A.X is extremely low even in the irradiated (DNA damaged) embryos at the 1-cell stage (Yukawa et al., BBRC 358:578-584, 2007). Therefore, we have deleted the description relating to gammaH2A.X.

Minor comments

1. The figure legends are not very informative. Inclusion of the details would be a welcome addition for the reader.

We are rather wondering that the current legends for figures are too long. We are afraid that the addition of more details would make them more redundant.

2. Nucleoli are not functional until the 8-cell stage in mouse embryos. Until this time these structures are referred to as nucleolar precursor bodies.

"Nucleolus" has been changed to "nucleolar precursor body" in the revised manuscript.

3. Pg 19 Line 2. Capitalisation missing.

It has been corrected.

4. Pg 18 reference Frey et al., should not include Current Biology.

The sentence including "Frey et al." has been removed because of the reason described above.

5. Pg 21 Line 8 has a 'fullstop' erroneously at the end of the line.
It has been corrected.

Reviewer #3 (Comments to the Authors (Required)):

Summary:

In this manuscript, Kawamura et al. investigate the nuclear patterns of histone H3.1/2 and H3.3 variants in preimplantation embryo development, more specifically 1-cell stage embryos.

In agreement with previous studies, the authors find that in murine zygotes, H3.1 and H3.2 are asymmetrically distributed between maternal and paternal pronuclei at the pericentromeric chromatin (PCH), with an relative enrichment on maternal pronuclei.

In brief, the new observations here are that when the authors overexpress H3.1/2 in 1-cell stage embryos, the variants can accumulate at paternal PCH. This is accompanied by a DNA replication delay of the PCH, and is further associated with a cleavage delay starting from the 2-cell stage. However, the abnormal accumulation of H3.1/2 in paternal PCH did not give rise to detectable changes in H3K9me2/3 or H3K27me3. Notably, previous work had shown that H3.3 K27 trimethylation at the paternal PHC was important for normal development (Santenard et al., Nat Cell Biol 2010). Thus, the authors overexpressed H3.1/2K27R in one-cell stage embryos, which surprisingly rescued the developmental delay present upon H3.1/2 overexpression. Based on these result they suggest that the K27 residue of H3.1/2 is important for mediating the effects observed upon H3.1/2 overexpression.

The manuscript does address an interesting issue of importance for developmental biology and epigenetics. However, at this stage the observations are not sufficiently solid and sometimes apparently inconsistent. The interpretation of the results is unclear and premature, since the authors do not provide direct evidence to support the fact that H3.1/2K27me3 is responsible for the PCH replication delay and developmental defects. There is also a need for them to revisit the background literature on the topic.

In conclusion, the proposed manuscript needs significant work to be considered for publication both experimentally and in the analysis to increase the quality of the data.

The conclusions are rather speculative and the implications unclear. Some potential issues should also be addressed more carefully, and the authors should provide quantification of the results (since many conclusions are only supported by representative images). A list of suggestions is detailed below.

Major points:

- Throughout the study the authors state that they have measured chromatin incorporation of H3 variants. However, they have performed IF for total H3.1/2, H3.3 or H3.1/2/3-FLAG (with the exception of Suppl. Fig. S4), which does not allow them to distinguish between chromatin-bound and free histones (Fig. 1, 2B, 3, 6, 7 and Suppl. Fig. S1, S2, S5B). Hence, the authors effectively describe nuclear patterns of H3.1/2 or H3.3, but not their incorporation into chromatin, and this should be stated more clearly in the text. Most importantly, the evidence provided does not allow to infer that H3.1/2 incorporation at the paternal PCH is responsible for the delay.

The reviewer is concerned about the possibility that the signals of H3 variants in the immunocytochemistry are derived from free histones in the nuclei but not chromatin-bound ones. However, we focus on the localization of H3 variants in the peripheral region of pronuclei but not whole nucleus: the free histones should be uniformly distributed in the nucleus.

Furthermore, we conducted the immunocytochemistry using wash-away method (Hajkova et al., Science 329: 78-82, 2010) in which the embryos had been treated with Triton-X100 before fixation to remove free histones in the nucleoplasm. These results were shown in Supplementary Fig. S4 and described in Page 9: Line 21 to Page 10: Line 7 in the original manuscript. In these sentences, we clearly described as “These results suggested that the detected histones are deposited in the chromatin.” I do not understand why the reviewer neglected these results and description about them.

- Similarly, the role of the K27 residue upon H3.1/2 overexpression in mediating DNA replication delay is unclear. Additional experiments directly demonstrating that incorporation of H3.1/2K27R in overexpressing embryos can prevent the replication delay should be performed.

Fig. 7C clearly shows that the incorporation of H3.1K27R or H3.2K27R did not affect the development. Conducting additional experiments which the reviewer suggested will take time and we are afraid that the lag of publication makes out of time in this field: it has already passed long time since I have submitted this manuscript to JCB and then to LSA.

Furthermore, this mutation ablates not only methylation, but also acetylation of K27 (and may also impact neighbouring PTMs). This point should be stated more clearly. This is especially critical since the authors state their results 'strongly suggested that the H3K27me3 modification on H3.1/2 (H3.1/2K27me3) was the determining factor for the delay in DNA replication and subsequent developmental failure' (page 16, lines 13-15). This is an important conclusion if true, but to make a strong argument this should be supported by further experiments showing that the effect is directly mediated by K27 trimethylation.

We have conducted an experiment to check the localization of H3K27ac in the perinucleolar region by immunocytochemistry. The results showed that K27ac is absent from perinucleolar region in 1-cell stage embryos. Therefore, H3K27ac is not related to the results of experiments using H3.1/2K27R. The results of experiments for H3K27ac have been added as a Supplementary Fig. S6 and described in the text (page 16, line 11-14).

- Many of the conclusions brought forward by the authors are only supported by representative IF images (Figs. 1, 3, 6 and Suppl. Figs. S2, S4 and S5) without quantifications of the signal or observed patterns at the maternal/paternal PHC or summary statistics. The authors should provide quantification and statistics to better corroborate their findings. Furthermore, error bars representing the standard deviation between experiments should be included to show variability in all bar plots (Fig. 4, 7C, Suppl. Fig S3, S6A) of IF signal (Fig. 5A, D).

It is too hard to quantify the signal intensity in the perinucleolar regions. The differences of signal intensities in these regions have been shown by the representative images in a number of papers. In the data representing by %, including error bars would be inappropriate.

- The authors state 'maternal and paternal perinucleolar regions were enriched in H3.1/2 with K9me2/3 and H3.3 with K27me3, respectively' (page 5, lines 17-18). However, in the images shown here, no clear enrichment of the variant is detectable at the paternal PHC (Fig. 1A, anti-H3.3, Fig. 3, neither no injection, nor H3.3 OE, Fig. 7A, Suppl. Fig. S4), although this has been previously described in the literature (Santenard et al., Nat Cell Biol 2010, Fig. 5a, b, Liu et al., EMBO J 2020, Fig. 1A PN5). How do the authors explain this discrepancy?

I suppose that the reviewer misunderstood the results. The signals in all of Fig. 1A, Fig. 3 and etc., which the reviewer mentioned, were detected by the antibody against H3.3.

Since the amount of H3.3 is larger in the nucleoplasm than the perinucleolar region, its signal is weak in the perinucleolar region. However, the signals in Fig. 6 which were detected by the antibody against H3K27me3 is stronger in the perinucleolar regions. These are obvious when the images stained with the antibodies against H3.3 and H3K27me3 are compared in Fig. 7A.

- Suppl. Fig. S6 is not described in the Results and is only introduced in the Discussion. It is not clear what the purpose of the data shown there is and how it fits with the rest of the manuscript.

As reviewer suggested, Supplementary Fig. S6 has been removed in the revised version.

- The model in Fig. 8 and many conclusions are largely speculative and not directly supported by the evidence presented in this manuscript. The authors did not investigate changes of neither chromatin structure nor the activity of lysine methyltransferases and they did not provide any data informing on their link to DNA replication timing. While some speculation can be stimulating here it seems that the basis for speculation is too weak, the model and the discussion of the data presented should be revised to show how it actually links to the authors' findings.

As reviewer suggested, the model in Fig. 8 is not based on the firm direct evidence. However, we suppose that a line of indirect evidence of several experiments supports it and that the presenting the model would be helpful for readers to easily understand the conclusion of this manuscript.

Other points:

- More details about the imaging system should be provided in the Methods section
The information of confocal laser scanning microscope was missing in the original manuscript. It has been added in the revised manuscript (page 24, line 24 to page 25, line 1).

- In Fig. 1C, nucleoli and chromocenters are not clearly distinguishable in the DAPI panel
We have conducted immunocytochemistry with the H3.1/3.2 antibody over again to obtain fine images in which nucleoli and chromocenters are clearly distinguishable, and the images of Fig. 1C has been replaced by new ones in the revised manuscript.

- In Fig. 2B, the y-axis label of the plot should be fixed. Also, are the p-values reported in the legend corrected for multiple testing?

The y-axis label has been added in Fig. 2B. In Fig. 2B, individual comparisons were performed and it would be good for detecting statistically significant differences in this

experiment.

The anti-FLAG staining (top panel) also does not show perinucleolar enrichment, this is in contrast with other images taken at the same stage with the anti-H3.1/2 or anti-H3.3 antibodies, why? How do the authors interpret this? Adding a DAPI panel for reference would also be helpful to display the chromatin density in these cells.

It is known that the addition of FLAG tag sometimes changes the intranuclear localization of the proteins. We suppose that it happened in this experiment.

- In several of the figures (Fig. 4A, 5A and Suppl. Fig. S3, S5A) the authors state they performed 'chi-squared or Fisher's exact test'. The authors should better clarify which test they performed (or why they performed two tests, if this is the case)

The reason why Fisher's exact test was performed has been described in the legends for Figure 4, 5 and Supplementary Fig. S3, S5.

- The authors state that the 'distributions of H3K9me2 and H3K9me3 modifications in the paternal pronuclei of H3.1- and H3.2-OEs did not differ from the control' (page 14, lines 19-20). However, there is an apparent reduction of H3K9me3 in all overexpressing embryos compared to no injection at both maternal and paternal pronuclei, and it appears to be less enriched at maternal PHC specifically in H3.1- and H3.2-OEs (Fig. 6).
We do not agree with this comment except for female PN in no injected embryos stained with anti-H3K9me2. This panel has been replaced with a representative one in which the signal is comparable with other samples.

- In Suppl. Fig. S2A, no H3.1/2 signal is detected at 15hpi embryo, whereas H3.1/2 is already present in the 11hpi Aphi (-) embryos in Suppl. Fig. S2B. How do the authors explain this?

At the beginning of the second paragraph of the results section (page 6, line 20-22 in the original manuscript), we clearly explain that H3.1/2 became visible when the confocal laser scanning microscope detector gain was enhanced (Fig. 1B), although it was not detected in the initial observation in which laser power was set at a standard level to compare the signal intensities among the embryos at the various stages of preimplantation development (Fig. 1A). At the observation in Fig. S2B, the laser power was increased to detect H3.1/2 signal. To avoid the confusion, we have added the description to explain it in the legend for Supplementary Figure. S2.

- The number of embryos injected per experiment should be specified in Suppl. Fig. S4

The numbers of embryos analyzed in experiments of Supplementary Fig. S4 have been described in the legend.

- The statement that 'H3.1 is enriched in the nucleolus during the latter stages of chromatin replication, whereas H3.3 localizes during early-stage chromatin replication (Clément et al., 2018); this suggests that H3 variants play a role in the regulation of DNA replication' (page 11, lines 5-8) should be rephrased to accurately reflect the findings of the study cited, which shows the H3 variant distribution correlates with replication timing but doesn't make claims as to a causal impact.

We agree with the reviewer's claim. We have deleted the description of "Clément et al., 2018".

- The statement that 'initiation of DNA replication in the nucleoplasmic regions was delayed for <2hpi and was completed by 10hpi' (page 13, lines 4-6) is not supported by the evidence presented, and should be rephrased.

We suppose that we can conclude those delay from the results of Fig. 5C, D.

- Similarly, the sentence 'PRC2 ... is not functional in the maternal perinucleolar region' (page 20, lines 8-10) is inaccurate: PRC2 is functional, but inhibited by the presence of HP1b at this region (Burton et al., Nat Cell Biol, 2020)

As the reviewer suggested, we have rewritten this part (page 20, line 3-5 in the revised manuscript).

- The authors state page 20, line 18, 'H3K27me3 is associated with facultative heterochromatin, H3.3 is associated with euchromatin (Hake et al., 2006, Hake and Allis, 2006); we therefore hypothesize that H3.3K27me3 forms heterochromatin with a loose structure, relative to H3.1/2K27me3'. However, H3.3 has also been shown to be present at pericentromeric and telomeric chromatin in embryonic stem cells and MEFs (Goldberg et al., Cell 2010 and Drané et al., Gen & Dev 2010, respectively). This shows an example of the need for the authors to re-examine the background literature.

We have rewritten this sentence in the revised version (page 20, line 12-13), as the reviewer suggested.

- The authors discuss the incorporation efficiency of H3.1/2 at 1-cell stage embryos (page 21, lines 17-20) and link it to the expression levels of the variants (Fig. 2) and of Caf1b (one of the homologues of one of the subunits of the CAF-1 complex which deposits

H3.1/2 on chromatin) which they measure by RT-PCR but do not show data for. The data supporting this statement should be included.

Since the data of CAF-1 component expression would be included in the future paper, I would not like to show them in this manuscript. Therefore, I show them for only reviewing process (the figure is added at the end of Supplemental Figures).

June 8, 2021

RE: Life Science Alliance Manuscript #LSA-2021-01102-TR

Dr. Fugaku Aoki
University of Tokyo
Kashiwanoha 5-1-5
Kashiwa 277-8562
Japan

Dear Dr. Aoki,

Thank you for submitting your revised manuscript entitled "Asymmetrical deposition and modification of histone H3 variants is essential for zygote development". We would be happy to publish your paper in Life Science Alliance pending final revisions necessary to meet our formatting guidelines. Please also clarify the following points:

- Was the data in Fig S6 showing a lack of H3-K27-Ac in the perinucleolar region obtained in a H3.1/2 overexpression scenario?
- Regarding, the FLAG tag changing intranuclear organization. Perhaps the FLAG tagged histones should not be used at all in this case.

- please add ORCID ID for the corresponding author-you should have received instructions on how to do so
- please add a Category for your manuscript in our system
- please upload your main and supplementary figures as single files
- please upload your main manuscript text as an editable doc file
- titles in the system and the manuscript file do not match. Please correct accordingly
- please add an Author Contributions section to your main manuscript text
- please add a conflict of interest statement to your main manuscript text
- please use the [10 author names, et al.] format in your references (i.e. limit the author names to the first 10)
- please add a statement indicating approval for performing the animal experiments
- there are Chinese characters in Figure 2 that should be removed

FIGURE CHECKS:

- Please add scale bars for Figures 1B, C; 5C; S6

A. FINAL FILES:

B. MANUSCRIPT ORGANIZATION AND FORMATTING:

Thank you for this interesting contribution, we look forward to publishing your paper in Life Science

Alliance.

Sincerely,

June 15, 2021

RE: Life Science Alliance Manuscript #LSA-2021-01102-TRR

Dr. Fugaku Aoki
University of Tokyo
Kashiwanoha 5-1-5
Kashiwa 277-8562
Japan

Dear Dr. Aoki,

Thank you for submitting your Research Article entitled "Asymmetrical deposition and modification of histone H3 variants is essential for zygote development". It is a pleasure to let you know that your manuscript is now accepted for publication in Life Science Alliance. Congratulations on this interesting work.

DISTRIBUTION OF MATERIALS:

Again, congratulations on a very nice paper. I hope you found the review process to be constructive and are pleased with how the manuscript was handled editorially. We look forward to future exciting submissions from your lab.

Sincerely,
